# Expressivity of Emergent Languages is a Trade-off between Contextual Complexity and Unpredictability

**Shangmin Guo**[†,*]**, Yi Ren**[‡]**, Kory Mathewson**[§]**, Simon Kirby**[†]**, Stefano V. Albrecht**[†]**, Kenny Smith**[†]

[†]University of Edinburgh, [‡]University of British Columbia, [§]DeepMind

## Abstract

Researchers are using deep learning models to explore the emergence of language in various language games, where agents interact and develop an emergent language to solve tasks. We focus on the factors that determine the *expressivity* of emergent languages, which reflects the amount of information about input spaces those languages are capable of encoding. We measure the expressivity of emergent languages based on the generalisation performance across different games, and demonstrate that the expressivity of emergent languages is a trade-off between the complexity and unpredictability of the context those languages emerged from. Another contribution of this work is the discovery of message type collapse, i.e. the number of unique messages is less than that of inputs. We also show that using the contrastive loss proposed by Chen et al. (2020) can alleviate this problem.

## 1 Introduction

Language games were first introduced by Wittgenstein (1954) to explore the meanings of language utterances. Instantiating this concept with the signalling game design from Lewis (1969) enables linguists to explore the emergence of linguistic structure (Kirby, 2001; Kirby & Hurford, 2002) where artificial languages are represented as symbolic systems. The success of deep learning (DL) models on complicated cognitive tasks (Krizhevsky et al., 2012; LeCun et al., 2015; Silver et al., 2016) then inspired researchers to apply DL-based models to language games to investigate the agents' ability to invent communication protocols without preset rules (e.g. Lee et al., 2018; Lazaridou et al., 2018).

In the existing works (e.g. Lee et al., 2018; Lazaridou et al., 2018; Graesser et al., 2019; Zhang et al., 2019; Yuan et al., 2020), there are usually two types of agents, *speakers* that emit messages based on their observations (i.e. input target objects) and *listeners* that receive messages and act accordingly, as illustrated in Figure 1. Based on the goals for listeners, we can categorise most of the games into the following three types: 1) *referential* games in which listeners need to select the target object observed by the speaker among a set of candidate objects (candidates for short) (e.g. Lazaridou et al., 2017; Ren et al., 2020); 2) *reconstruction* games in which listeners need to reconstruct the speaker's observation (e.g. Chaabouni et al., 2020; Kharitonov et al., 2020); and 3) *navigation* games in which listeners need to go to the location specified by speakers (e.g. Lowe et al., 2017; Kajić et al., 2020). We focus on referential games, illustrated in Figure 1a, which have been well investigated in both the linguistic community (e.g. Kirby et al., 2014; Winters et al., 2018) and emergent communication community (e.g. Lazaridou et al., 2018; Li & Bowling, 2019).

It is reasonable to assume that listeners in referential games can differentiate the target object from the other distractors in the context as long as the speaker's message encodes some unique feature of the target. Therefore, the speaker's messages must convey a different amount of information for listeners to complete their task in games where candidates are similar in different degrees. For example, to distinguish two very similar candidates, e.g. blue wide-stripped shirt and blue narrow-stripped shirt, it requires more information about the target to be encoded by the speaker's message than to distinguish a shirt from a cup. Experiments with human participants show that emergent languages are sensitive to the requirements of the communicative tasks for which they are used, with languages developing in which only the necessary discriminatory information is encoded (Winters et al., 2018).

---

*Correspondence author, e-mail address: s.guo@ed.ac.uk

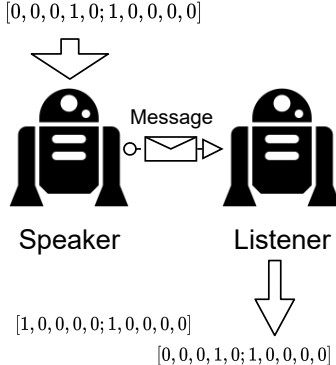

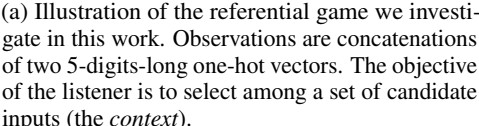

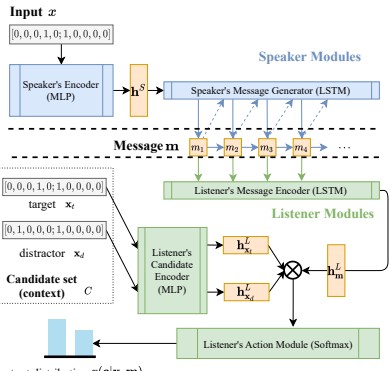

(a) Illustration of the referential game we investigate in this work. Observations are concatenations of two 5-digits-long one-hot vectors. The objective of the listener is to select among a set of candidate inputs (the *context*).

(b) Illustration of the agent architecture we implement for referential games. Blue modules are components of the speaker, and green modules belong to the listener. Orange and grey indicate the vectors generated by agents and input samples, respectively.

Figure 1: Diagram of referential game and the architecture of our agents.

Following Kirby et al. (2015), we refer to such ability to encode information about input space (e.g. structure, dimensionality, distribution of samples) as **expressivity**. In this work, we explore the factors that influence expressivity in the framework of DL-based referential games, and we argue that it is important to understand these factors such that agents could develop languages that are effective enough for completing various tasks.

Our contribution in this work is threefold. First, we propose and verify a hypothesis about the determining factors of expressivity of emergent languages under the DL-based framework. Following Winters et al. (2018), we show that context size (i.e. the number of candidates the listener must select among when interpreting the speaker's message) influence both the *complexity* and *unpredictability* of that context. Complexity refers to the similarity between the items in the context: in order for the listener to select the target, more information needs to be encoded in the speaker's signal if the context contains more similar distractors for a given target. Unpredictability refers to the extent to which the information that needs to be encoded by the speaker is stable across contexts in successive episodes of communication: in games where the information required by the listener differs from trial (training epoch for DL agents) to trial, the speaker needs to encode more information on *every* trial in order to be sure to encode the information the listener needs.[1] As we show in Section 3.1, complexity and unpredictability are affected differently by the context size: as context size increases, complexity increases but unpredictability decreases. Therefore, we propose and verify the following hypothesis about the determining factors of expressivity:

[HYPOTHESIS] *The expressivity of emergent languages is a trade-off between the complexity and the unpredictability of context in language games.* [1]

Our second contribution is a novel measure of expressivity based on partial ordering of languages in terms of their generalisation performances across tasks. Although expressivity is related to the amount of information encoded in a language, we illustrate that mutual information (MI) is not an appropriate measurement for expressivity in Section 3.2. Considering that one of our objectives is to facilitate a universally useful emergent language, we propose to measure expressivity of a language based on the generalisation performance of listening agents trained with that language across different games. Since it is more challenging to quantitatively and directly measure expressivity, we focus on the *partial ordering* between languages in this first step towards understanding expressivity.

Our final contribution is the discovery of message type collapse, i.e. the number of unique messages is significantly lower than the size of input space in relatively easy games, which can lead to highly ambiguous emergent languages. To overcome the technical limitations imposed by GPU memory size on large-scale referential games, we introduce the contrastive loss proposed by Chen et al. (2020) in referential games. While comparing the behaviour of the contrastive loss and the loss function used in previous works (e.g Havrylov & Titov, 2017; Li & Bowling, 2019; Ren et al., 2020; Guo et al., 2020), we find that this contrastive loss can greatly alleviate the collapse of message types, leading to more disambiguous and potentially more expressive emergent languages.

---

[1]Context, complexity, and unpredictability are defined and discussed in depth in Section 3.

## 2 Games, Agents and Learning Methods

### 2.1 Game Environment and Architecture of Agents

**Environment Components** As shown in Figure 1b, formally speaking, the environment in our referential games consists of: 1) *input space* $\mathcal{X}$ from which the speaker's observations are drawn, which consists of $10,000$ possible inputs[2] where each $x \in \mathcal{X}$ is a concatenation of 4 one-hot vectors whose length is 10; 2) *message space* $\mathcal{M}$ consists of $10^6$ 6-tokens-long sequences $m$ (called as a *message type*) where the size of the token inventory is 10; note that the tokens are initially meaningless, and meaning emerges through the negotiation of a communication protocol between speakers and listeners; 3) *candidate set* $C$ which contains a target input $x_t$ and $|C| - 1$ distractors $\{x_{d_1}, x_{d_2}, \ldots, x_{d_{|C|-1}}\}$ sampled uniformly at random from $\mathcal{X}$ without replacement.

**Agents** The architecture of the agents we implemented is also illustrated in Figure 1b: 4) *speaker $S$* consists of both a multi layer perceptron (MLP) for encoding the 40-digits-long target input $x_t$ onto a 256-digits-long embedding $h^S$, and a decoder for generating a message $m$ based on long-short term memory (LSTM) network (Hochreiter & Schmidhuber, 1997). 5) *listener $L$* is constituted by an LSTM-based sequence encoder which encodes $m$ into a 256-digits-long embedding $h^L(m)$, and an MLP for encoding the inputs which can be denoted by $f_{cenc}^L(\cdot)$.

**Definition of an emergent language** Following e.g. Ren et al. (2020), we also define an emergent language $\mathcal{L}$ as a mapping function from the input space $\mathcal{X}$ to the message space $\mathcal{M}$, i.e. $\mathcal{L} : \mathcal{X} \mapsto \mathcal{M}$, thus it can be represented as $\{(x_i, \mathcal{L}(x_i)) | \forall x_i \in \mathcal{X}\}$. Note that $\mathcal{L}$ could be a non-injective mapping function, i.e. it is possible for two different inputs to be mapped to an identical message thus the number of message types might be less than the number of unique inputs.

### 2.2 Learning methods of agents

In the previous works (e.g. Havrylov & Titov, 2017), the action module of $L$ is usually a softmax function which produces a categorical distribution over the candidates based on the distance between $h^L(m)$ and embeddings of each candidate ($f_{cenc}^L(x_i)$) obtained through listener's candidate encoder $f_{cenc}^L(\cdot)$. By comparing the loss function used in Havrylov & Titov (2017) and the contrastive loss function proposed by Hadsell et al. (2006), we find that the aim of both functions is the same, i.e. to make the distance between positive pairs ($x_t$ and $m$) closer while keeping the negative pairs ($x_d$ and $m$) as far apart as possible. To effectively validate our hypothesis, we need to make the whole input space $\mathcal{X}$ as candidate set $C$ in which case the batch is too large to be stored in GPU memory (since the space complexity is $\mathcal{O}(|\mathcal{X}|^2)$). Therefore, to overcome this issue, we adapt the contrastive learning method proposed by Chen et al. (2020) to referential games. Suppose that we have a batch of input samples $B = \{x_1, x_2, \ldots, x_i, \ldots, x_{|B|}\}$ and the corresponding messages from the speaker are $\{m_{B_1}, m_{B_2}, \ldots, m_{B_i}, \ldots, m_{B_{|B|}}\}$ (note that it is possible that $m_{B_i} = m_{B_j}$ for different $i, j$) which would be further encoded into $\{h^L_{m_{B_1}}, h^L_{m_{B_2}}, \ldots, h^L_{m_{B_i}}, \ldots, h^L_{m_{B_{|B|}}}\}$ by the listener $L$. Then the loss function for a sample $x_i$ is the defined on a softmax function:

$$\ell_i = -\log \frac{\exp\left(h^L_{m_i}{}^\top \cdot f_{cenc}^L(x_i)\right)}{\sum_{j=1}^{|B|} \exp\left(h^L_{m_i}{}^\top \cdot f_{cenc}^L(x_j)\right)}. \tag{1}$$

In this way, the number of candidates $|C|$ is exactly the batch size $|B|$ during training. More details about the comparison between referential loss and this contrastive loss can be found in Appendix A.

As for updating the parameters, we use the Adam algorithm introduced by Kingma & Ba (2015), and the learning rate is set to $10^{-4}$. To allow the gradients being propagated through the discrete channel to overcome the sampling issue of messages, we apply the Gumbel-Softmax trick proposed by Jang et al. (2020), and the temperature hyper-parameter $\tau$ is set to $1.0$.

Our implementation[3] of games is based on the framework *EGG* developed by Kharitonov et al. (2019). The experiments across 6 random seeds, 18 source games, 11 target games took $4,216$ hours in total, on Nvidia Tesla P100. Definition of "source/target games" are given in the following Section 4.2.

---

[2]According to the results from Chaabouni et al. (2020), an input space containing $10,000$ possible inputs is large enough for neural models to achieve $> 99\%$ generalisation accuracy.

[3]Codes are released at https://github.com/uoe-agents/Expressivity-of-Emergent-Languages.

# 3 CONTEXT AND EXPRESSIVITY

## 3.1 CONTEXT, AND ITS COMPLEXITY AND UNPREDICTABILITY IN DIFFERENT GAMES

**Context** Communication between humans always takes place w.r.t. some context (Winters et al., 2018). For humans, that context includes: 1) features of the environment that can be directly perceived by the dialogue participants and used for interpreting an utterance (e.g. "cup" is interpreted as an object in the immediate physical context); and 2) information that cannot be directly perceived but can affect the interpretation of utterances (e.g. using and interpreting "cup" assumes some shared knowledge of what counts as a "cup"; once a particular cup has been established as salient in particular dialogue, further utterances of "the cup" or "it" will be interpreted as referring to that specific cup).

Inspired by this conclusion from pragmatics, in the DL-based language games, we define *context* as the *supervision information* involved in the calculation of losses, i.e. the context of a specific target $\boldsymbol{x}_t$, denoted as $\mathcal{C}(\boldsymbol{x}_t)$, is the space of samples involved in the calculation of loss. As for in referential games, the cross entropy loss is calculated based on the distance between the message embedding $\boldsymbol{h}^L(\boldsymbol{m}_{B_i})$ and candidate embedding $f^L_{cenc}(\boldsymbol{x}_i)$ where $\boldsymbol{x}_i \in C$, thus $\mathcal{C}(\boldsymbol{x}_t) = C \subseteq \mathcal{X}$ in referential games.

**Complexity of context** We assume that the goal of communication is to distinguish a target object from other possibilities in the context (as defined above). It therefore follows that the similarity of distractors in the context to the target influences the communicative precision required, and that greater precision is required to distinguish the target from a more similar distractor. For example, it is relatively easy in natural language to distinguish e.g. a cat from a table (a relatively general label like "cat" or "animal" would suffice), but harder to make fine-grained distinctions between very similar objects e.g. a Scottish Fold Cat and an American Shorthair Cat (a specialist vocabulary or a lengthy description is necessary).

Following the assumption (described verbally above) that a context which contains more similar objects makes the game harder because there are fewer unique features that suffice to distinguish the target from the distractors, we first define a neighbour set in $k$-th degree of a given target $\boldsymbol{x}_t$ as $\mathcal{N}_k(\boldsymbol{x}_t) = \{\boldsymbol{x}_j : d(\boldsymbol{x}_t, \boldsymbol{x}_j) \leqslant k\}$ where $d$ is a distance function that can properly capture the similarity between inputs, e.g Hamming distance in our setting. The complexity of $\mathcal{C}(\boldsymbol{x}_t)$ is then defined as the expectation of the probability that $\mathcal{C}(\boldsymbol{x}_t)$ contains an object from $\mathcal{N}_k(\boldsymbol{x}_t)$, i.e.

$$\mathbb{E}_{\boldsymbol{x}_t}\left[g\left(\mathcal{C}(\boldsymbol{x}_t), \mathcal{N}_k(\boldsymbol{x}_t)\right)\right] \text{ ,where } g\left(\mathcal{C}(\boldsymbol{x}_t), \mathcal{N}_k(\boldsymbol{x}_t)\right) = \begin{cases} 1, & \text{if } \exists \boldsymbol{x}_i \in \mathcal{C}(\boldsymbol{x}_t) \text{ s.t. } \boldsymbol{x}_i \in \mathcal{N}_k(\boldsymbol{x}_t) \\ 0, & \text{otherwise} \end{cases} \quad (2)$$

In our referential games, since the sampling procedure is independent Bernoulli without replacement, the value of the above expectation is then $1 - \left(\frac{|\mathcal{X}| - 1 - |\mathcal{N}_k(\boldsymbol{x}_t)|}{|\mathcal{X}| - 1}\right)^{|C|}$ which is a monotonically increasing function w.r.t $|C|$ and a fixed $k$. That said, larger contexts are more complex since they are more likely to include items which are very similar to the target.

**Unpredictability of context** Our notion of unpredictability comes from experimental work with human participants, e.g. Winters et al. (2018). Suppose the aim is to distinguish a striped t-shirt from a number of distractors, and there are two sequences of context: 1) three runs of games where distractors are all cups; 2) three runs of games where distractors are a cup, a plain t-shirt, and a pencil. In the first sequence of games, participants would become certain that "t-shirt" is enough for distinguishing the target, whereas in the second sequence participants would learn that a more overspecified utterance (e.g. "striped t-shirt") is necessary to guarantee comprehension after a failure on the trial involving the plain t-shirt distractor. That is, the context in the first sequence is more predictable than the second. Winters et al. (2018) show that human participants are sensitive to this kind of unpredictability, and adapt their communicative strategies accordingly.

In DL-based games, we refer to the context at the $e$-th trial, i.e. the $e$-th training epoch, of a target $\boldsymbol{x}_t$ as $\mathcal{C}^e(\boldsymbol{x}_t)$. Following the above example, the *unpredictability* of context is then defined as the

proportion of $\mathcal{C}^{e+1}(\boldsymbol{x}_t)$ that are not from $\mathcal{C}^e(\boldsymbol{x}_t)$, i.e.

$$\mathbb{E}_{\boldsymbol{x}_t} \left[ \frac{1}{|\mathcal{C}^{e+1}(\boldsymbol{x}_t)|} \sum_{\boldsymbol{x}_j \in \mathcal{C}^{e+1}(\boldsymbol{x}_t)} \mathbb{I}\left(\boldsymbol{x}_j \notin \mathcal{C}^e(\boldsymbol{x}_t)\right) \right] \tag{3}$$

In our referential games, since the sampling procedure is independent Bernoulli without replacement, the proportion of objects not from $\mathcal{C}^e(\boldsymbol{x}_t)$ in $\mathcal{C}^{e+1}(\boldsymbol{x}_t)$ is then simply $1 - \frac{|\mathcal{C}^e(\boldsymbol{x}_t)|}{|\mathcal{X}|} = 1 - \frac{|C|}{|\mathcal{X}|}$ (i.e. smaller contexts are more unpredictable, since contexts on successive trials are more likely to differ in their composition).

More details about the above discussion are provided in Appendix B.

## 3.2 Measuring Expressivity in Language Games

Following the intuition that the different types of games require messages to convey different amount of information about input space $\mathcal{X}$, MI may seem to be a good candidate measure for expressivity. However, we found that even for two emergent languages that are both deterministic and bijective mapping functions, i.e. they have the same MI value $\log(|\mathcal{X}|)$, their expressivity can still be different. Further discussion about MI can be found in Appendix D.

Since it is more challenging to directly quantify the expressivity of emergent languages, we instead focus on the partial orders between them in this work. Considering that the purpose of the emergent communication is to complete tasks, we therefore propose to measure the partial order between expressivity based on game performance. This leads us to a common practice in the transfer learning community (e.g. Pan & Yang, 2009), i.e. train models on source domain but evaluate them on different target domains. Inspired by this practice, here we propose a definition of the partial order between the expressivity of two emergent languages $\mathcal{L}_A$ and $\mathcal{L}_B$, i.e. $\mathfrak{E}_{\mathcal{L}_A}$ and $\mathfrak{E}_{\mathcal{L}_B}$:

**Definition 3.1** ($\mathfrak{E}_{\mathcal{L}_A}^{\mathcal{G}} > \mathfrak{E}_{\mathcal{L}_B}^{\mathcal{G}}$). Let $g$ be an instance of language game, and $\mathcal{G}$ be a set of language game instances. Then, given $\mathcal{G}$, if the generalisation performance of $\mathcal{L}_A$ is better than $\mathcal{L}_B$ on some language games $\mathcal{G}'$ with statistically significant difference, i.e. $\forall g \in \mathcal{G}'$, $\mathfrak{P}_{\mathcal{L}_A}^g > \mathfrak{P}_{\mathcal{L}_B}^g$, where $\varnothing \subset \mathcal{G}' \subseteq \mathcal{G}$ and $\mathfrak{P}$ denotes the generalisation performance, while there is no statistically significant difference between the converged generalisation performance of $\mathcal{L}_A$ and $\mathcal{L}_B$ on the remaining games, i.e. $\mathfrak{P}_{\mathcal{L}_A}^{g'} \approx \mathfrak{P}_{\mathcal{L}_B}^{g'} \forall g' \in \mathcal{G} \backslash \mathcal{G}'$ [4], we then say that expressivity of $\mathcal{L}_A$ is higher than $\mathcal{L}_B$ on $\mathcal{G}$, i.e. $\mathfrak{E}_{\mathcal{L}_A}^{\mathcal{G}} > \mathfrak{E}_{\mathcal{L}_B}^{\mathcal{G}}$. The metric of generalisation performance could be varied according to different types of games.

Briefly speaking, given a set of games, if one language performs better than the other on some games while they perform approximately the same on the remaining games, we then say the expressivity of that language is higher than the other. Following this definition, there are several possible relationships between the expressivity of two languages $\mathcal{L}_A$ and $\mathcal{L}_B$: 1) $\mathfrak{E}_{\mathcal{L}_A}^{\mathcal{G}} > \mathfrak{E}_{\mathcal{L}_B}^{\mathcal{G}}$, defined as above; 2) $\mathfrak{E}_{\mathcal{L}_A}^{\mathcal{G}} = \mathfrak{E}_{\mathcal{L}_B}^{\mathcal{G}}$, if $\mathfrak{P}_{\mathcal{L}_A}^g \approx \mathfrak{P}_{\mathcal{L}_B}^g$, $\forall g \in \mathcal{G}$; 3) $\mathfrak{E}_{\mathcal{L}_B}^{\mathcal{G}} > \mathfrak{E}_{\mathcal{L}_A}^{\mathcal{G}}$, defined as above; 4) $\mathfrak{E}_{\mathcal{L}_A}^{\mathcal{G}}$ is *incomparable* to $\mathfrak{E}_{\mathcal{L}_B}^{\mathcal{G}}$ ($\mathfrak{E}_{\mathcal{L}_A} \gtrless \mathfrak{E}_{\mathcal{L}_A}$), if propositions $\exists g' \in \mathcal{G}$, $s.t.$ $\mathfrak{P}_{\mathcal{L}_A}^{g'} > \mathfrak{P}_{\mathcal{L}_B}^{g'}$ and $\exists g'' \in \mathcal{G}$, $s.t.$ $\mathfrak{P}_{\mathcal{L}_B}^{g''} > \mathfrak{P}_{\mathcal{L}_A}^{g''}$ are both true.

## 4 Predictions and Experiment Designs

### 4.1 Predictions From Our Hypothesis

As we illustrated in Section 3.1, higher complexity indicates more similar distractors to targets, thus requires messages conveying more fine-grained features of inputs. This leads to the following prediction:

1. higher complexity in games with same contextual unpredictability can facilitate emergent languages having higher expressivity;

---

[4]Note that $\approx$ in this work means that there is no statistically significant difference between means of two sequences.

Meanwhile, following the conclusion from Winters et al. (2018) that higher contextual unpredictability leads to higher signal autonomy, i.e. more comprehensive information about input space is encoded in the emergent languages, we then make the following prediction:

2. higher unpredictability in games having same contextual complexity can facilitate emergent languages having higher expressivity;

Following the above two predictions, both higher complexity and higher unpredictability can unilaterally lead to better expressivity. However, with an input space whose size is finite, changing context size always affects the two factors in an inverse way, which leads to our Hypothesis 1 and gives the following prediction:

3. if the size of $\mathcal{X}$ is fixed across games, a language which emerges in a game with moderate complexity and unpredictability should have the highest expressivity.

## 4.2 LANGUAGE TRANSFER EXPERIMENT

To verify the above three predictions and thus our hypothesis, we evaluate the generalisation performance of languages emerging in various games following the Procedure 1. Briefly speaking, we first train a pair of speaker and listener to complete a referential game, i.e. a "source game" $g_s$, and obtain an emergent language. We then train a new listener in another referential game, i.e. a "target game" $g_t$, with *only* 90% of the input-message pairs, and use the game performance of this listener with the remaining 10% pairs as the generalisation performance of $g_s$ on $g_t$.

---

**Procedure 1:** Procedure for the language emergence and transfer experiment

---

Input: A set of source game $\mathcal{G}_s$, a set of target game $\mathcal{G}_t$

**for** *every game $g_s^i$ in $\mathcal{G}_s$* **do**

    1. initialise a new speaker and listener for $g_s^i$, and train them to play $g_s^i$ with the whole $\mathcal{X}$;

    2. after the agents converge on $g_s^i$, record $\mathcal{L} = \{(\boldsymbol{x}, \boldsymbol{m}) | \boldsymbol{x} \in \mathcal{X}\}$;

    3. randomly shuffle and split $\mathcal{L}$ into 2 disjoint sets $\mathcal{L}_{train}$ and $\mathcal{L}_{test}$ s.t. $|\mathcal{L}_{train}| = 90\% \cdot |\mathcal{L}|$;

    4. **for** *every game $g_t^j$ in $\mathcal{G}_t$* **do**

        1. initialise a new listener for $g_t^j$;

        2. train the listener with $\mathcal{L}_{train}$ to complete $g_t^j$;

        3. record the *accuracy* of listener on $\mathcal{L}_{test}$ as the **generalisation performance of $g_s^i$ on** $g_t^j$;

    **end**

**end**

---

Since our hypothesis concerns the size of $C$, we propose to experiment on a series of referential games with varying candidate set size $|C|$ ("referX" where X= $|C|$). Meanwhile, as these referential games have both different complexity and unpredictability, we also implement a series of referential games with different $|C|$ but the batches are fixed during training such that there is no unpredictability of the context ("referXf" where X= $|C|$). To sum up, in our experiment, $\mathcal{G}_s =$ {"refer2/f", "refer10/f", "refer100/f", "refer1000/f", "refer2500/f", "refer5000/f", "refer7500/f", "refer10000"}[5]. Since language games with fixed context are not good simulation of human communication , we keep only the ones with varying batches from $\mathcal{G}_s$ in $\mathcal{G}_t$.

As a naming convention to allow us to refer easily to the factors that influence a language's expressivity, we refer to emergent languages by the parameters of their source game, e.g. 'refer10'[6] represents a language which emerges in referential games whose $|C| = 10$ and context varies across epochs. More details about the naming of games and languages can be found in Appendix E.1.

## 5 RESULTS & DISCUSSION

To alleviate the effect from randomness as much as possible, all the following results are obtained with 6 different random seeds. All values are averaged across the seeds, and the corresponding standard deviation (SD) are also provided (either in this section or appendices).

---

[5]There is no "refer10000f" since no unpredictability is possible when $|B| = |C| = 10,000 = |\mathcal{X}|$

[6]"referX" is a game while 'referX' is an emergent language from it.

## 5.1 HIGHER COMPLEXITY + SAME UNPREDICTABILITY → HIGHER EXPRESSIVITY

To verify the Prediction 1, we compare the generalisation performance of languages from fixed context, i.e. the unpredictability of source games' context are the same, on $\mathcal{G}_t$, and the results are shown in Figure 2. Languages which emerge in source games whose contexts have higher complexity consistently perform better on all games than those which emerge in source games with lower-complexity contexts, since all lines keep growing with the increase of values on $x$-axis (i.e. complexity). That is, the generalisation performance is a *monotonically increasing* function of $|C|$ if there is no unpredictability in the context, which matches with the prediction of our hypothesis.

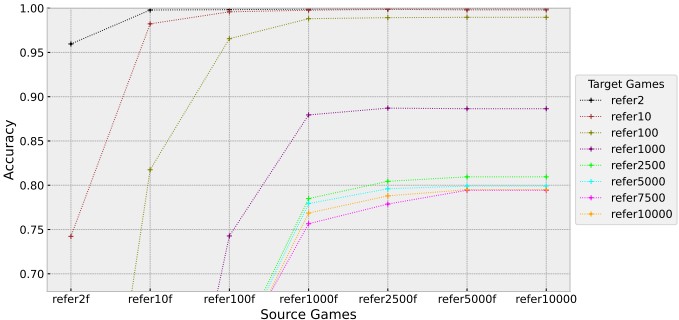

Figure 2: Generalisation performance of languages from fixed context with varying complexity. The $x$-axis indicates the source games, and $y$ is the accuracy. Lines in different colours represent the generalisation performance on different target games. We plot only the means of the converged performance to keep the figure readable (SD values are provided in Table 3 in Appendix E.2), and remove 'refer7500f' from the x-axis since it is a mix of $|C| = 7,500$ and $|C| = 2,500$.

## 5.2 HIGHER UNPREDICTABILITY + SAME COMPLEXITY → HIGHER EXPRESSIVITY

To verify the Prediction 2, we compare the difference between the generalisation performance of language with varying context ('referX') and their corresponding fixed-context twins ('referXf'). As indicated by that all points in Figure 3 are positive, languages from context with varying objects always perform better on all target games. Since all "referXf" has lower unpredictability of context than the corresponding "referX" games, this indicates that higher unpredictability can unilaterally lead to better generalisation performance of emergent language across various target games, which matches with the prediction of our hypothesis.

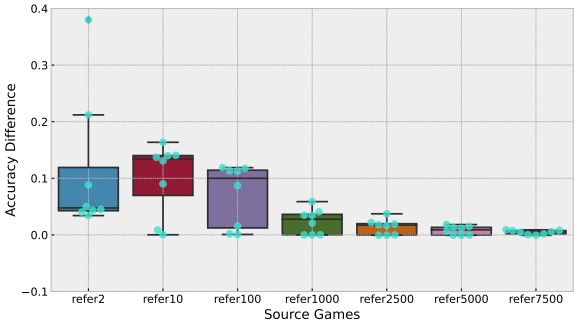

Figure 3: Generalisation performance gaps between languages from varying context and their corresponding fixed-context twins on all target games. The $x$-axis indicates the types of source games, and the $y$-axis indicates the distribution of difference between the generalisation performance of 'referX' and 'referXf' on all target games (i.e. 'referX' - 'referXf'). Each point represent the difference on a specific kind of target game, and the boxes show the corresponding values of medians, 25th-percentile, and 75-th percentile. "refer10000" is not plotted since it doesn't have a fixed-context twin game.

### 5.3 EXPRESSIVITY IS A TRADE-OFF BETWEEN CONTEXTUAL COMPLEXITY AND UNPREDICTABILITY

To verify the Prediction 3, we compare performance for all source games with varying context on all target games, and the results are shown in Figure 4 (see also Appendix E.3 for more details).

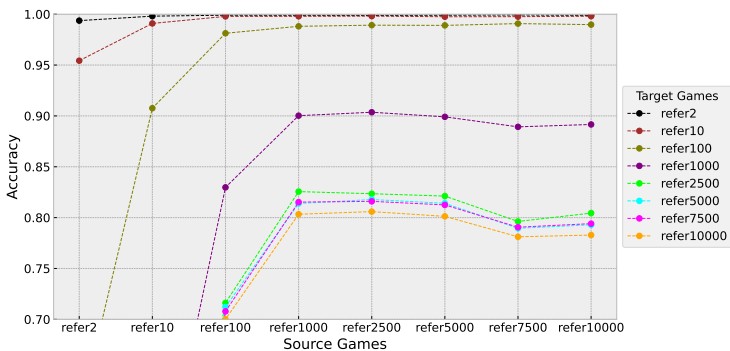

Figure 4: Generalisation performance of language from different source games on all target games. Each line represents a specific kind of *target game*. The $x$-axis indicates the type of source games, and the $y$-axis is the accuracy on target games. To keep the figure readable, we remove the error bars, and instead provide the corresponding SD values in Table 4. We limit the range of $y$-axis to $[0.7, 1.0]$ to emphasise the difference between languages on complex games, refer to Figure 8 for more values.

As shown in Figure 4, on whichever type of game, the peaks of all curves are always among "refer1000", "refer2500", or "refer5000", which means that the best expressivity always emerges in these 3 types of games. Considering that the complexity of context in referential game is a *monotonically increasing* function of the $\frac{|C|}{|\mathcal{X}|}$ while the contextual unpredictability is a *monotonically decreasing* of it, the results shown in Figure 4 support our hypothesis, i.e. the expressivity is indeed a trade-off between the contextual complexity and unpredictability.

To be specific, although context in "refer2", "refer10", and "refer100" have high unpredictability, their complexity is low thus the languages have lower expressivity. On the other hand, context of "refer7500" and "refer10000" have high complexity, but expressivity of neither language is the highest due to the high predictability of the context in these games. This supports our argument that unpredictability of context incentivises the agents to encode more information into the messages such that the messages could overcome the uncertainty of contexts and be interpreted in various contexts. Therefore, the three languages ('refer1000', 'refer2500', and 'refer5000') from context with both moderate complexity and unpredictability perform better than the others. To make it more straightforward to compare the expressivity between these languages, we also provide another view of the above results in Figure 9 in Appendix E.3, which shows that 'refer1000' $\not\gtrless$ 'refer2500' $\approx$ 'refer5000' > 'refer7500' $\approx$ 'refer10000' > 'refer100' > 'refer10' > 'refer2'.

To eliminate the possibility that the above phenomena are caused by the hyperparameters we used in our games, we also run the language transfer experiment with varying communication channel capacity and agent capacity (i.e. the size of hidden layers). As shown in the figures in Appendix F, both 'refer1000' and 'refer2500' perform better than 'refer10000' across varying hyperparameters, which shows that the trade-off phenomenon is consistent over game settings and agent capacity.

### 5.4 THE COLLAPSE OF MESSAGE TYPES

As we use a contrastive loss instead of the conventional referential loss applied by (e.g. Havrylov & Titov, 2017; Ren et al., 2020), we also compare the behaviours of both loss functions on $\mathcal{G}'_s = \mathcal{G}'_t = \{$"refer2", "refer10", "refer100"$\}$. We discover that the number of message types is some times greatly less than the size of input space. We refer such phenomenon as "message type collapse", and the mappings from several different inputs to a same message type as "degeneracy components".

While comparing the two loss functions, we observe that contrastive loss is more efficient on removing degenerate components in large-scale games (where $|C| \geqslant 100$), and can better avoid the collapse of

the number of message types. To be specific, as shown in Figure 5a, the numbers of message types collapses to less than $500$ on games with conventional referential loss. They, however, would only decrease to roughly $5,000$ on 'refer10' and 'refer1000' with contrastive loss as shown in Figure 5b. The contrastive loss can also make the training of referential game more stable, as illustrated by the smaller variance in Figure 5b.

Our results indicate that it is necessary to establish a referential game with candidate sets containing enough many distractors to guarantee that the emergent language will be relatively unambiguous. As far as we know, the existing works do not investigate the effects of the size of candidate sets, e.g. Lee et al. (2018) set $|C| = 2$, Li & Bowling (2019) set $|C| = 5$, and $|C| \leqslant 20$ is used by Lazaridou et al. (2018). Given our results, our heuristic is that $|C|$ should be greater than either $1,000$ for large input spaces or $\frac{1}{10}|\mathcal{X}|$ for smaller ones.

We also explore more about the degeneracy degree (as well as structure degree) of the mappings constituting different languages, and the results and discussion are provided in Appendix C.

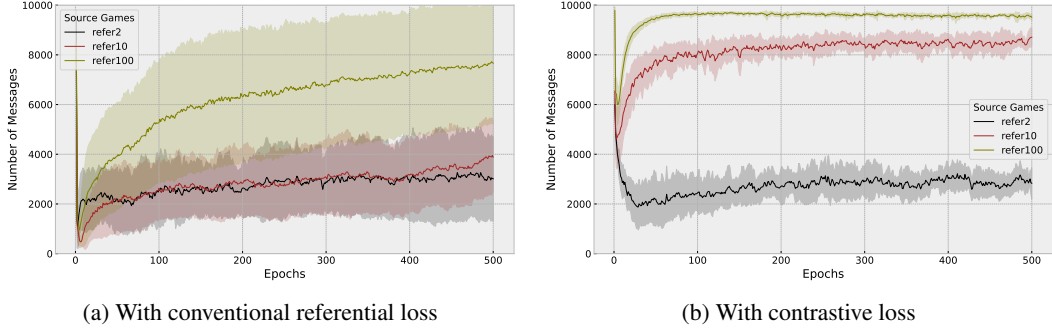

(a) With conventional referential loss                            (b) With contrastive loss

Figure 5: How number of message types change over training epochs with different loss functions. The lines are the means over 6 runs and shadow areas indicate the corresponding variance.

## 6 CONCLUSION & FUTURE WORK

To our knowledge, this is the first work exploring the *expressivity* of emergent languages from language games played by DL-based agents that have different context size. We first proposed a hypothesis about the factors influencing the expressivity of emergent languages, and defined a partial ordering between expressivity of different emergent languages based on the generalisation performance across a set of games. Through a series of experiments on referential games with varying size of context, we then validated the three predictions of our hypothesis, which furthers indicates that expressivity is indeed a trade-off between the complexity and unpredictability of context in language games. To overcome the memory inefficiency issue caused by the conventional referential loss function, we introduced a contrastive loss into our implementation. In the comparison between this contrastive loss and the referential loss, we found the message type collapse problem, and also showed that using this contrastive loss can alleviate it.

Since this is a first step towards defining and understanding the expressivity of emergent languages, there are still some questions unanswered by this work. For example, some properties of our emergent languages are surprising given the simplicity of some of our referential games. In the simplest setting ($|C| = 2$), we see emergent languages with more than $2,000$ message types, whereas an optimal strategy requires a language consists of only $40$ unique messages (since there are only $40$ different values for all properties in $\mathcal{X}$ and two items in the context always differ on at least one property). Thus, it shows that further investigation about how redundancy helps during the communication between agents is necessary. More importantly, we lack understanding of the structure of mappings constituting a language, especially when the languages are generated or encoded by neural networks. We believe that the expressivity of languages still needs further exploration in order to facilitate languages that are universally effective across various tasks.

ACKNOWLEDGEMENT

This project has also received funding from the European Research Council (ERC) under the European Union's Horizon 2020 research and innovation programme under grant agreement No. 681942, held by K. S.

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

# A  MORE DETAILS ABOUT COMPARISON BETWEEN DIFFERENT LOSS FUNCTIONS

In this section, we give more details about the comparison between the referential loss used by previous works and the contrastive loss we use in Section 2.2.

Following the same notations used in Section 2.2, the referential loss used by (e.g. Havrylov & Titov, 2017) can be written as:

$$p(c_i|\boldsymbol{x}_i, \boldsymbol{m}) \propto \exp E(\boldsymbol{h}_{\boldsymbol{m}}^L, f_{cenc}^L(\boldsymbol{x}_i)) = \exp\left(\boldsymbol{h}_{\boldsymbol{m}}^{L^\top} \cdot f_{cenc}^L(\boldsymbol{x}_i)\right) \tag{4}$$

where $\boldsymbol{h}_{\boldsymbol{m}}^L$ is the embedding for the message $\boldsymbol{m}$ obtained from the sequence encoder of listener $L$, and $f_{cenc}^L(\cdot)$ is the candidate encoding function of $L$. From the above equation, we can see that if a candidate has an embedding closer to message $\boldsymbol{m}$, the probability mass for predicting it would then be higher. Hence, the predicated probability mass is inversely proportional to the distances, e.g. the dot product in the above equation, between the embeddings of $\boldsymbol{m}$ and all candidates.

The contrastive loss function proposed by Chen et al. (2020) with $\tau = 1$ is given as follow:

$$\ell_{i,j} = -\frac{\exp\left(\text{sim}(\boldsymbol{x}_i, \boldsymbol{x}_j)\right)}{\sum_k \mathbb{I}_{[k \neq i]} \exp\left(\text{sim}(\boldsymbol{x}_i, \boldsymbol{x}_k)\right)} \tag{5}$$

where $\mathbb{I}$ is an indicator function. By comparing the above loss function with Equation 4, we find that the aim of them is the same, i.e. make the distance between positive pairs (targets and their corresponding messages) closer while keeping the embeddings for negative pairs (distractors and the messages) as far apart as possible. While negative sampling has been deeply investigated in the self-supervised learning (SSL) community, (e.g. Chen et al., 2020; Grill et al., 2020), it has not been discussed in the emergent communication community yet, thus we simply use all randomly sampled distractors as negative examples. As mentioned in Section 2.2, we need to make the whole input space as candidate set, thus the above contrastive loss which has lower space complexity is preferred.

In the following, we give a bit further derivation to show that Equation 1 is computationally equivalent to Equation 5. By substituting $k$ in the denominator of Equation 5 with $j$, we can see that:

$$\begin{aligned}
\ell_{i,j} &= -\log \frac{\exp\left(\text{sim}(\boldsymbol{x}_i, \boldsymbol{x}_j)\right)}{\sum_j \mathbb{I}_{[j \neq i]} \exp\left(\text{sim}(\boldsymbol{x}_i, \boldsymbol{x}_k)\right)} \\
&= \log \frac{\sum_j \mathbb{I}_{[j \neq i]} \exp\left(\text{sim}(\boldsymbol{x}_i, \boldsymbol{x}_k)\right)}{\exp\left(\text{sim}(\boldsymbol{x}_i, \boldsymbol{x}_j)\right)} \\
&= \log \frac{\sum_j \mathbb{I}_{[j \neq i]} \exp\left(\text{sim}(\boldsymbol{x}_i, \boldsymbol{x}_k)\right)}{\exp\left(\text{sim}(\boldsymbol{x}_i, \boldsymbol{x}_j)\right)} + \underbrace{\log \frac{\exp\left(\text{sim}(\boldsymbol{x}_i, \boldsymbol{x}_i)\right)}{\exp\left(\text{sim}(\boldsymbol{x}_i, \boldsymbol{x}_i)\right)}}_{=0} \\
&= \log \frac{\sum_j \exp\left(\text{sim}(\boldsymbol{x}_i, \boldsymbol{x}_j)\right)}{\exp\left(\text{sim}(\boldsymbol{x}_i, \boldsymbol{x}_j)\right)} \\
&= -\log \frac{\exp\left(\text{sim}(\boldsymbol{x}_i, \boldsymbol{x}_j)\right)}{\sum_j \exp\left(\text{sim}(\boldsymbol{x}_i, \boldsymbol{x}_j)\right)}
\end{aligned} \tag{6}$$

# B  MORE DETAILS ABOUT CONTEXTUAL COMPLEXITY/UNPREDICTABILITY

In this section, we give more details about the definition of context and its complexity/unpredictability.

## B.1  CONTEXT

As defined in Section 3.1, the context for a target object $\boldsymbol{x}_t$ is the space of samples involved in the calculation of its loss. We know that the prediction of the listener is calculated based on the distance between the message received ($\boldsymbol{m}(\boldsymbol{x}_t)$) which is a function of the target input ($\boldsymbol{x}_t$), i.e. $\text{softmax}\left(d_1\left(\boldsymbol{m}(\boldsymbol{x}_t), \boldsymbol{x}_d\right)\right)$ where $\boldsymbol{x}_d$ is a distractor and $d_1$ is a distance function for embeddings. Cross entropy loss is then calculated based on $\text{softmax}$ function. Thus, we can also write the loss function as $\mathcal{L}(m(\boldsymbol{x}_t), \{\boldsymbol{x}_d\}; \theta)$. As shown in the equation, the calculation is based on samples from candidate set $C$. Therefore, in referential game, $\mathcal{C}(\boldsymbol{x}_t) = \{\boldsymbol{x}_t\} \cup \{\boldsymbol{x}_d\} = C \subseteq \mathcal{X} \subset \mathcal{O}$.

## B.2 COMPLEXITY OF CONTEXT

As defined in Section 3.1, the $k$-th degree neighbour set is $\mathcal{N}_k(\boldsymbol{x}_t) = \{x_j : d(\boldsymbol{x}_t, \boldsymbol{x}_j) \leqslant k\}$ where $k \in \mathbb{Z}^+$. As described in Section 2.1, there are 4 features in our synthetic data set and each of them can take 10 different values. Therefore, for a specific target $\boldsymbol{x}_t$, the $k$ in the definition of neighbour set can take only the following 4 values:

- $k = 1$: samples in $\mathcal{N}_1(\boldsymbol{x}_t)$ differ only in 1 feature from $\boldsymbol{x}_t$. For the only different feature, there are only 9 possible values a distractor can take. Thus, $|\mathcal{N}_1(\boldsymbol{x}_t)| = 4 \times 9 = 36$.

- $k = 2$: samples from $\mathcal{N}_1(\boldsymbol{x}_t)$ and samples that differ at exactly 2 features from $\boldsymbol{x}_t$. For the later case, there are first $\binom{4}{2} = 12$ possible combinations of such two features, and there are $9 \times 9 = 81$ possible samples for each combination. Thus, $|\mathcal{N}_2(\boldsymbol{x}_t)| = 12 \times 81 + 36 = 1008$.

- $k = 3$: samples from $\mathcal{N}_2(\boldsymbol{x}_t)$ and samples that differ at exactly 3 features from $\boldsymbol{x}_t$. For the later case, there are first $\binom{4}{3} = 4$ possible combinations of such three features, and there are $9 \times 9 \times 9 = 729$ possible samples for each combination. Thus, $|\mathcal{N}_3(\boldsymbol{x}_t)| = 4 \times 729 + 1008 = 3924$.

- $k = 4$: all samples in $\mathcal{X}$ except $\boldsymbol{x}_t$ are in this neighbour set, thus $|\mathcal{N}_4(\boldsymbol{x}_t)| = 10000 - 1 = 9999$

Since we use independent Bernoulli sampling, for each draw, the probability to get a sample from $\mathcal{N}_k(\boldsymbol{x}_t)$ is simply $\frac{\mathcal{N}_k(\boldsymbol{x}_t)}{|\mathcal{X}|-1}$ where $|\mathcal{X}| - 1$ is due to excluding $\boldsymbol{x}_t$ from $\mathcal{X}$. Meanwhile, since we define $\mathcal{C}(\boldsymbol{x}_t)$ as the candidate set in referential games. Therefore, for $|\mathcal{C}(\boldsymbol{x}_t)| = |C|$ draws, the probability of having *at one* sample from $\mathcal{N}_k(\boldsymbol{x}_t)$ is:

$$1 - \left(1 - \frac{|\mathcal{N}_k(\boldsymbol{x}_t)|}{|\mathcal{X}| - 1}\right)^{|C|} = 1 - \left(\frac{|\mathcal{X}| - 1 - |\mathcal{N}_k(\boldsymbol{x}_t)|}{|\mathcal{X}| - 1}\right)^{|C|}.$$

## B.3 UNPREDICTABILITY OF CONTEXT

In this section, we first derive our measure of unpredictability following Winters et al. (2018). In their experiments, the input space consists of 16 referents. There are 2 different attributes, *colour* and *shape*, and each attribute has 4 different values. By setting the size of candidate set to 4, they define the following two types of context:

- **shape-different**: all the 3 different distractors have identical colour to the target, but different shapes. Since there are only 4 shapes and candidate set size is also 4, in this setting, the distractors for a given target will never change during the games. This is exactly the same as our "fixed-batch" games defined in Appendix E.1.

- **mixed**: all the 3 different distractors are sampled uniformly at random from the input space, thus they may have different shapes as well as different colours to the target. In this case, the distractors for a given target vary across different trials during the games.

By comparing the two above cases, it is straightforward to see that it is more possible to see different distractors for a given target in the games with mixed context. Following this heuristic, we proposed to measure the unpredictability of context by the expected chance of observing different distractors for targets across training epochs, which further leads us to the definition given in Equation 3. Note that there is also complexity (under our definition) involved in the above two types of context, we focus only on the variability part of the context in our definition of unpredictability since the complexity part has been taken into consideration by the neighbourhood set defined in Section 3.1.

We then give further explanation about Equation 3 and its simplified form. Since we use Bernoulli sampling for each individual object in candidate set, the overall sampling of candidate set is thus a binomial procedure. As the probability of getting a sample from $\mathcal{C}^e(\boldsymbol{x}_t)$ in $\mathcal{X}$ is $\frac{|\mathcal{C}^e(\boldsymbol{x}_t)|}{|\mathcal{X}|}$, the overall proportion of objects *from* $\mathcal{C}^e(\boldsymbol{x}_t)$ in $\mathcal{C}^{e+1}(\boldsymbol{x}_t)$ is the same, i.e. $\frac{|\mathcal{C}^e(\boldsymbol{x}_t)|}{|\mathcal{X}|}$. Therefore, the probability of the complementary event, i.e. getting an object *not from* $\mathcal{C}^e(\boldsymbol{x}_t)$ in $\mathcal{C}^{e+1}(\boldsymbol{x}_t)$, is $1 - \frac{|\mathcal{C}^e(\boldsymbol{x}_t)|}{|\mathcal{X}|}$.

## C   More Results and Discussion on Expressivity

In this section, we provide two more intermediate properties of emergent languages that are also influenced by complexity and unpredictability of context, and thua may influence the expressivity.

### C.1   Degree of Degeneracy Affects but Does Not Determine the Expressivity

Inspired by Kirby et al. (2015), the first intermediate factor we explore is the *degree of degeneracy* (i.e. the degree of ambiguity) of the languages. As a language is defined as a mapping function $\mathcal{L} : \mathcal{X} \mapsto \mathcal{M}$, the degree of its degeneracy could be measured by the number of distinct message types.

As shown in the Table 1, in referential languages, the degeneracy degree decreases rapidly with the increase of the contextual complexity of the source games, and reaches a ceiling around $|D| \geqslant 1,000$. There is a positive correlation between the number of message types and expressivity for the languages with low complexity, which indicates that degree of degeneracy could be an intermediate factor that is correlated with the expressivity of these languages. However, the fact that the average number of message types on refer10000 is higher than refer1000, while the expressivity of 'refer1000' is higher than 'refer10000', i.e. emergent languages having lower level of degeneracy does not necessarily have higher expressivity, indicates that degeneracy degree of one language cannot determine its expressivity.

| Game | refer2 | refer10 | refer100 | refer1000 | refer2500 | refer5000 | refer7500 | refer10000 |
|---|---|---|---|---|---|---|---|---|
| Mean | 2872.18 | 8565.69 | 9547.58 | 9801.60 | 9839.33 | 9841.27 | 9840.34 | 9856.96 |
| 25th-percentile | 2629.0 | 8325.0 | 9501.5 | 9778.75 | 9825.0 | 9837.5 | 9819.0 | 9845.0 |
| 75th-percentile | 3195.0 | 8795.5 | 9639.25 | 9823.25 | 9854.25 | 9864.25 | 9866.0 | 9871.0 |

Table 1: Mean, 25th-percentile, and 75th-percentile of the distributions of message types numbers from different source game.

### C.2   Structure Degree of Mappings of Different Languages

Since the degenerate degree is not the only factor determining expressivity, another factor to be considered is how the mappings of a language $\mathcal{L}$ are structured after the training procedure of agents on source games. According to the cardinality of the mappings of $\mathcal{L}$, we can divide them into two disjoint sets: i) *one-to-one* mappings, i.e. an input is mapped to a unique message; and ii) *many-to-one* mappings (a.k.a degenerate components), i.e. several inputs are mapped to the same message type. Between this two types, the structure of one-to-one mappings has been explored under the view of compositionality in the previous works, e.g. (Brighton & Kirby, 2006; Andreas, 2019). However, as illustrated by Chaabouni et al. (2020), for emergent languages from DL-based agents, a good generalisability does *not* require a good compositionality, which means that measurements for compositionality are not good predictors for expressivity either. In this work, we focus on only the *many-to-one* mappings.

Given the objectives of games, for the degenerate components in an emergent language, a message $m$ being mapped to several closely-related inputs could imply that it captures common features among these inputs. Suppose that several inputs $X_K = \{x_1, \ldots, x_K\}$ are mapped to the same message $m$ in a converged language $\mathcal{L}$, we hypothesise that *the average distances between every pair of inputs*, i.e. $\frac{1}{|X_K|} \sum_{i,j \in \{1,\ldots,K\}} d(x_i, x_j)$, would become smaller after training, compared with their random initial values. Therefore, we compare the averaged distances between meanings of degenerate components as well as the frequency of degenerate components at the beginning and end of training. To make the plots more readable, we only plot the statistics about the top-10 most frequent ambiguous message types, where the frequency of a message type is defined as the number of inputs that are mapped to it in a languages. Results on 'refer1000' (one of the most expressive language in our experiments) are shown in Figure 6.

By comparing the data of beginning and end of training in Figure 6b, we can easily observe that *the averaged distance* decreased throughout the training, which indicates that the degenerate components actually became more although not fully structured. Another observation we have limited understanding of is that 'refer1000' becomes more ambiguous after training (shown by that the red bars

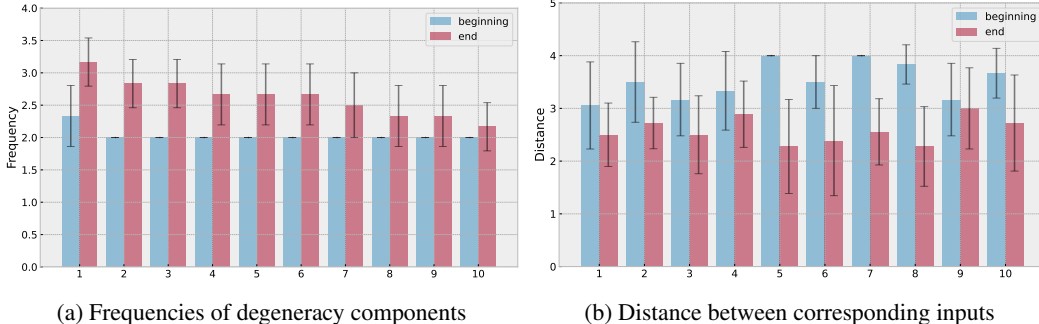

(a) Frequencies of degeneracy components

(b) Distance between corresponding inputs

Figure 6: Statistics about frequency of *Top-10* most frequent message type and the average Hamming distances between their meanings at the beginning and end of training on 'refer1000'. The bars indicate the frequency and the Hamming distance, and the black lines indicate the corresponding SD.

representing the statistics at the end of training are lower than the bars representing the statistics at the beginning of training), although its accuracy does improve to almost $100\%$. However, limited by the scope of this work, we will leave further exploration of it to future works.

## D  MUTUAL INFORMATION IS INSUFFICIENT

In Section 3.2, we mentioned that MI cannot completely reflect the expressivity of an emergent language. Here, we provide more detailed explanations. On one hand, a higher MI value does not necessarily mean more task-relevant information, e.g. encoding colour in messages is unnecessary for tasks involving only the shapes of inputs although it provides more information. On the other hand, not only the determinacy of the mappings matters for generalising them, but also the structures, e.g. it would be easier for humans to extrapolate compositional languages to novel concepts than holistic ones although they are both unambiguous and therefore have the same MI. In the following paragraphs, we will give more reasoning and empirical evidence to support our claim.

Following the mapping function definition of emergent languages illustrated in Section 2.1, suppose that we get a message type collection $M \subset \mathcal{M}$ in some training epoch, the calculation of the mutual information between $M$ and $\mathcal{X}$ is given as follows:

$$I(M|X)$$

$$= \sum_{\boldsymbol{x} \in X} \sum_{\boldsymbol{m} \in M} p(\boldsymbol{x}, \boldsymbol{m}) \log \frac{p(\boldsymbol{x}, \boldsymbol{m})}{p(\boldsymbol{x}) p(\boldsymbol{m})}$$

$$\overset{p(\boldsymbol{x}, \boldsymbol{m}) = \frac{1}{|\mathcal{X}|}}{=} \sum_{\boldsymbol{x} \in \mathcal{X}} \sum_{\boldsymbol{m} \in \mathcal{M}'} \frac{1}{|\mathcal{X}|} \log \frac{p(\boldsymbol{x}, \boldsymbol{m})}{p(\boldsymbol{x}) p(\boldsymbol{m})}$$

$$= \frac{1}{|\mathcal{X}|} \sum_{\boldsymbol{x} \in X} \sum_{\boldsymbol{m} \in M} \log \frac{p(\boldsymbol{x}, \boldsymbol{m})}{p(\boldsymbol{x}) p(\boldsymbol{m})}$$

$$= \frac{1}{|\mathcal{X}|} \sum_{\boldsymbol{x} \in X} \sum_{\boldsymbol{m} \in M} (\log p(\boldsymbol{x}, \boldsymbol{m}) - \log p(\boldsymbol{x}) - \log p(\boldsymbol{m}))$$

$$\overset{p(\boldsymbol{x}) = \frac{1}{|\mathcal{X}|}}{=} \frac{1}{N} \sum_{\boldsymbol{x} \in X} \sum_{\boldsymbol{m} \in M} \left( \log \frac{1}{|\mathcal{X}|} - \log \frac{1}{|\mathcal{X}|} - \log p(\boldsymbol{m}) \right)$$

$$= \frac{1}{|\mathcal{X}|} \sum_{\boldsymbol{x} \in \mathcal{X}} \sum_{\boldsymbol{m} \in M} (- \log p(\boldsymbol{m}))$$

$$= - \sum_{\boldsymbol{m} \in M} \log p(\boldsymbol{m})$$

$$= - \sum_{\boldsymbol{m} \in M} \log \frac{|\mathcal{L}(\boldsymbol{m})|}{|M|}$$

$$= - \sum_{\boldsymbol{m} \in M} \left( \log |\mathcal{L}(\boldsymbol{m})| - \log |M| \right)$$

$$= \sum_{\boldsymbol{m} \in M} \left( \log |M| - \log |\mathcal{L}(\boldsymbol{m})| \right)$$

where $p(\boldsymbol{x}) = \frac{1}{|\mathcal{X}|}$ as $\boldsymbol{x}$ follows a uniform distribution over the input space, $p(\boldsymbol{x}, \boldsymbol{m}) = \frac{1}{|\mathcal{X}|}$ since every $\boldsymbol{x}$ is mapped to a message, and $|\mathcal{L}(\boldsymbol{m})|$ is the appearance number of $\boldsymbol{m}$ in one language.

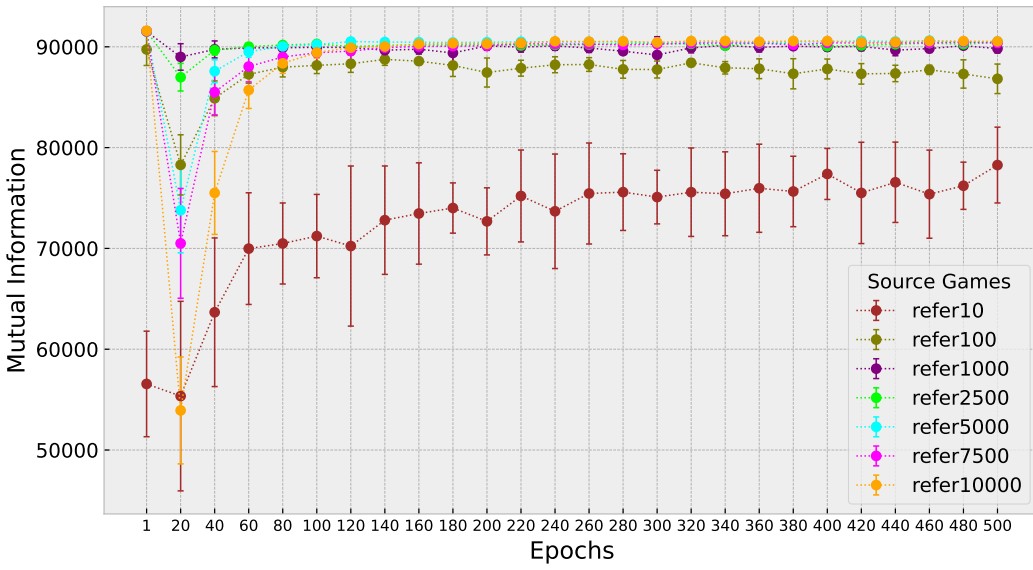

Figure 7: Mutual information curves over training epochs in different types of games. The converged values are provided in Table 2

By observing the above equation, it is straightforward to see that the MI value is decided by the number of message types, i.e. the degeneracy degree of one language. Therefore, if two emergent languages have approximately the same number of messages types (as shown in Table 1 and degeneracy degree, the MI values for these two languages should also be the same. However, as we illustrate in Section C.1 that degree of degeneracy cannot determine the expressivity of emergent languages, we can then deduce that MI cannot fully measure the expressivity of emergent languages.

To provide more empirical evidence to support our claim, we tracked the MI values over training epochs on a subset of $\mathcal{G}_s$, and the results are shown in the following Figure 7. As we can see, there is no statistically significant difference between the MI values of 'referX' (where X$\geqslant 1,000$) while it has been shown in Section 5.3 that their expressivity is different.

| Game | refer10 | refer100 | refer1000 | refer2500 | refer5000 | refer7500 | refer10000 |
|------|---------|----------|-----------|-----------|-----------|-----------|------------|
| Mean | 78269.22 | 86823.66 | 89823.16 | 90447.69 | 90504.40 | 90378.89 | 90533.06 |
| SD | 3760.05 | 1464.69 | 443.12 | 213.76 | 412.37 | 306.85 | 139.42 |

Table 2: Converged MI from different source games. Both mean and standard deviation are given.

# E  LANGUAGE TRANSFER EXPERIMENT RESULTS

## E.1  NAMING OF GAMES AND LANGUAGES

In this section, we give more details about the naming of games and corresponding languages we used in Section 5 as follows:

- "referX": this refers to the referential games with candidate sets whose size is X (which itself is an integer), note that we use double quotation marks to indicate that this is a specific type of game. For example, "refer2500" refers to referential games with $2,500$ candidates for listeners.

- 'referX': this refers to a emergent languages from "referX", note that we use apostrophes here in order to differentiate it from the notations of the corresponding game. For example, 'refer2500' refers to emergent languages from referential games with $2,500$ candidates for listeners.

- "referXf": this refers to the referential games with candidate sets whose size is X, but the partition of batches is identical over epochs, thus giving the game a fixed context. For example, "refer2500f" refers to referential games with $2,500$ candidates for listeners, but the batches of data are kept identical through the whole training procedure.

- 'referXf': this refers to the emergent languages from "referXf". For example, 'refer2500f' refers to emergent languages from referential games with $2,500$ candidates for listeners with fixed context. Regarding the meaning of "fixed-batch" or "fixed context", we give an example below where the whole input space is $\{a, b, c, d\}$ and batch size is 2:
    - varying batch/context: $\{a, b\}, \{c, d\}$ is the partition for the 1st epoch; then $\{a, c\}, \{b, d\}$ is used for the 2nd epoch; and random partitions are used in the following epochs;
    - fixed batch/context: whichever epoch it is, $\{a, b\}, \{c, d\}$ is always used as the partition of context;

- "referX/f": this equals a "referX" and a "referXf", both of them have same complexity but different unpredictability. To be more specific, contextual unpredictability of "referX" is higher than "referXf".

- $\mathcal{G}_s$: the universal set of source games which is defined as {"refer2", "refer10", "refer100", "refer1000", "refer2500", "refer5000", "refer7500", "refer10000", "refer2f", "refer10f", "refer100f", "refer1000f", "refer2500f", "refer5000f", "refer7500f", }.

- $\mathcal{G}_t$: the universal set of target games which is defined as {"refer2", "refer10", "refer100", "refer1000", "refer2500", "refer5000", "refer7500", "refer10000"}.

## E.2 RAW RESULTS OF COMPLEXITY EXPERIMENT

In this section, we provide the raw data of Figure 2 in the following Table 3. Not that although "refer1000f" performs the best on "refer10" than all other languages, we argue this outlier doesn't influence conclusion of our prediction considering the simplicity of "refer10".

| Source | Statistics | refer2 | refer10 | refer100 | refer1000 | refer2500 | refer5000 | refer7500 | refer10000 |
|---|---|---|---|---|---|---|---|---|---|
| refer2f | mean | 0.9596 | 0.7423 | 0.2244 | 0.0322 | 0.0168 | 0.0163 | 0.0156 | 0.0163 |
| | $\sigma$ | 0.0236 | 0.1493 | 0.1308 | 0.0203 | 0.0103 | 0.0105 | 0.0111 | 0.0119 |
| refer10f | mean | 0.9979 | 0.9823 | 0.8175 | 0.3259 | 0.2028 | 0.1918 | 0.1909 | 0.1879 |
| | $\sigma$ | 0.0007 | 0.0044 | 0.0404 | 0.0518 | 0.0461 | 0.0446 | 0.0500 | 0.0418 |
| refer100f | mean | 0.9983 | 0.9958 | 0.9655 | 0.7428 | 0.5988 | 0.5933 | 0.5946 | 0.5871 |
| | $\sigma$ | 0.0010 | 0.0029 | 0.0106 | 0.0573 | 0.0738 | 0.0685 | 0.0719 | 0.0663 |
| refer1000f | mean | 0.9989 | 0.9977 | 0.9881 | 0.8794 | 0.7849 | 0.7793 | 0.7565 | 0.7686 |
| | $\sigma$ | 0.0008 | 0.0006 | 0.0032 | 0.0163 | 0.0250 | 0.0242 | 0.0664 | 0.0237 |
| refer2500f | mean | 0.9992 | 0.9987 | 0.9892 | 0.8871 | 0.8044 | 0.7960 | 0.7788 | 0.7880 |
| | $\sigma$ | 0.0008 | 0.0005 | 0.0032 | 0.0199 | 0.0259 | 0.0239 | 0.0267 | 0.0264 |
| refer5000f | mean | 0.9988 | 0.9980 | 0.9897 | 0.8916 | 0.8094 | 0.7980 | 0.7941 | 0.7928 |
| | $\sigma$ | 0.0007 | 0.0006 | 0.0023 | 0.0077 | 0.0099 | 0.0143 | 0.0127 | 0.0122 |
| refer10000 | mean | 0.9993 | 0.9980 | 0.9897 | 0.8864 | 0.8095 | 0.7991 | 0.7943 | 0.7950 |
| | $\sigma$ | 0.0005 | 0.0007 | 0.0021 | 0.0084 | 0.0142 | 0.0095 | 0.0123 | 0.0107 |

Table 3: Mean and standard deviation $\sigma$ of generalisation performance of languages from fixed-context on various target games.

### E.3 MORE RESULTS OF TRADE-OFF EXPERIMENT

In this section, we first provide the complete version of Figure 4, i.e. Figure 8. Since the performance of 'refer2', 'refer10', and 'refer100' are too bad on complex target games, it is very clear that their expressivity are much lower that the rest. Therefore, to emphasise the difference between the other languages, we limit the range of $y$-axis to $[0.7, 1.0]$ in Figure 8.

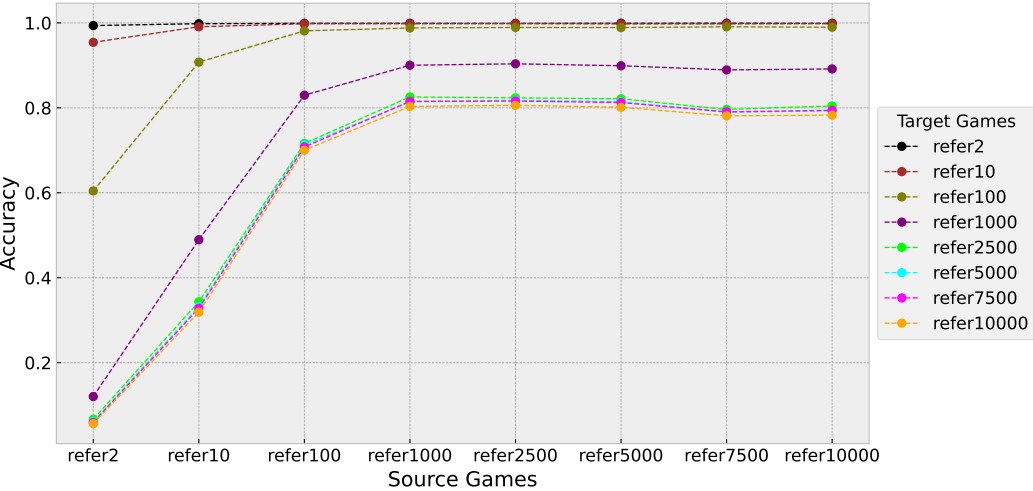

Figure 8: Same to Figure 4, but with wider range for $y$-axis.

We also illustrate the partial order between the expressivity of different languages in Figure 9. It is very clear that the 'refer2', 'refer10', and 'refer100' performs worse than the others on referential games with more candidates than their sources. Combining this fact with the fact that 'refer7500' and 'refer10000' perform worse than 'refer1000', 'refer2500', and 'refer5000' shows that increasing the size of candidate set $|C|$ would not always benefit the expressivity. As for 'refer1000', 'refer2500', and 'refer5000', there is no statistically significant difference between their performance on the simpler referential games, and they intersect on some games, thus their expressivity are either incomparable or approximately the same.

At last, we give the precise values of both Figure 9 and 4 in the following Table 4. All the results are obtained by multiple runs, thus both mean and standard deviation ($\sigma$) are given.

To better support our claim, we also provide $p$-values of testing the statistical significance between the generalisation performance of two different source games across all types of target games in the following Table 5.

## F VARYING THE CAPACITY OF CHANNEL AND NETWORK

To verify that our findings are robust to the capacity of the communication channel and agents, we also run some extra experiments on the following different configurations of referential games:

1. *larger channel capacity*: set the length of messages to $8$ and size of token inventory to $20$, thus the size of message space becomes $20^8$;

2. *larger agent capacity*: set the hidden layer size to $512$, thus the capacity of both speaker and listener become larger than the original setting ($256$);

3. *larger channel&agent capacity*: do the above two changes at the same time.

Since our key findings are about the generalisation performance between refer1000, refer2500, and refer10000, we just run these extra experiments on a game set $\mathcal{G}_{extra} = \{$'refer1000', 'refer2500', 'refer10000'$\}$ with multiple runs. The results of configuration 1, 2, and 3 are given in Figure 10, 11, and 12 respectively. It can be observed in all figures that 'refer1000' or

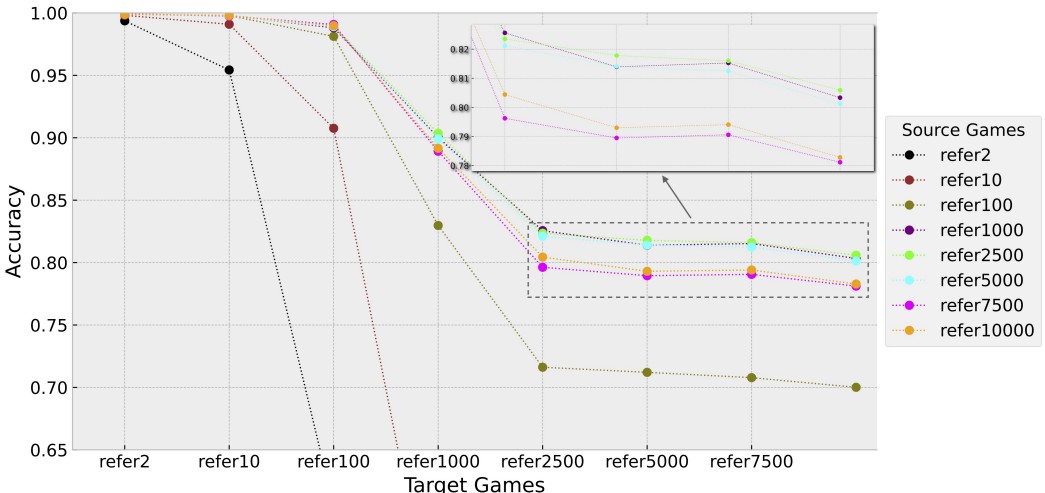

Figure 9: Generalisation performance of language from the same source game over different target games, where each line represents a *source game*. Since the plot would become not readable if we plot error bars here, we provide SD in Table 4 below. The x-axis indicates the type of target games, and the y-axis is the accuracy on target games. If a line lies above the other over all the target games, it means that the corresponding source language has a higher expressivity than the other. On the other hand, if two lines intersect, e.g. 'refer1000' and 'refer2500' above, then the expressivity of these languages is *incomparable*.

| Source | Statistics | refer2 | refer10 | refer100 | refer1000 | refer2500 | refer5000 | refer7500 | refer10000 |
|---|---|---|---|---|---|---|---|---|---|
| refer2 | mean | 0.9936 | 0.9543 | 0.6043 | 0.1203 | 0.0668 | 0.0616 | 0.0592 | 0.0565 |
| | $\sigma$ | 0.0014 | 0.0077 | 0.0394 | 0.0146 | 0.0105 | 0.0078 | 0.0090 | 0.0047 |
| refer10 | mean | 0.9980 | 0.9909 | 0.9076 | 0.4896 | 0.3434 | 0.3318 | 0.3282 | 0.3188 |
| | $\sigma$ | 0.0007 | 0.0036 | 0.0245 | 0.0833 | 0.0809 | 0.0801 | 0.0778 | 0.0752 |
| refer100 | mean | 0.9986 | 0.9978 | 0.9812 | 0.8298 | 0.7162 | 0.7121 | 0.7078 | 0.7001 |
| | $\sigma$ | 0.0007 | 0.0009 | 0.0035 | 0.0171 | 0.0299 | 0.0312 | 0.0293 | 0.0350 |
| refer1000 | mean | 0.9989 | 0.9979 | 0.9881 | 0.9003 | 0.8256 | 0.8139 | 0.8153 | 0.8033 |
| | $\sigma$ | 0.0007 | 0.0009 | 0.0015 | 0.0143 | 0.0151 | 0.0165 | 0.0226 | 0.0183 |
| refer2500 | mean | 0.9988 | 0.9981 | 0.9892 | 0.9036 | 0.8235 | 0.8178 | 0.8160 | 0.8059 |
| | $\sigma$ | 0.0006 | 0.0009 | 0.0012 | 0.0121 | 0.0099 | 0.0158 | 0.0148 | 0.0197 |
| refer5000 | mean | 0.9990 | 0.9975 | 0.9890 | 0.8991 | 0.8212 | 0.8141 | 0.8125 | 0.8013 |
| | $\sigma$ | 0.0004 | 0.0004 | 0.0022 | 0.0050 | 0.0140 | 0.0095 | 0.0117 | 0.0140 |
| refer7500 | mean | 0.9990 | 0.9975 | 0.9890 | 0.8893 | 0.7963 | 0.7895 | 0.7906 | 0.7812 |
| | $\sigma$ | 0.0005 | 0.0004 | 0.0034 | 0.0156 | 0.0206 | 0.0185 | 0.0223 | 0.0215 |
| refer10000 | mean | 0.9988 | 0.9980 | 0.9887 | 0.8916 | 0.8044 | 0.7930 | 0.7940 | 0.7828 |
| | $\sigma$ | 0.0007 | 0.0005 | 0.0023 | 0.0077 | 0.0099 | 0.0143 | 0.0127 | 0.0122 |

Table 4: Mean and standard deviation $\sigma$ of generalisation performance of language games on each other.

'refer2500' always perform better than 'refer10000' on all kinds of target games, i.e. 'refer1000' or 'refer2500' both have higher expressivity than 'refer10000' . Therefore, we believe the key observation from Figure 9 holds across configurations of communication channel and capacity of agents, which means that our conclusions are robust to the configuration of language games.

| Sources | refer2 | refer10 | refer100 | refer1000 | refer2500 | refer5000 | refer7500 | refer10000 |
|---|---|---|---|---|---|---|---|---|
| refer2 vs refer10 | **2.73e-06** | **5.27e-04** | **1.63e-06** | **1.27e-04** | **5.05e-04** | **5.92e-04** | **5.51e-04** | **5.56e-04** |
| refer2 vs refer100 | **7.44e-06** | **3.82e-04** | **6.70e-06** | **1.57e-13** | **3.78e-09** | **2.12e-08** | **6.69e-07** | **1.00e-07** |
| refer2 vs refer1000 | **4.62e-06** | **3.47e-04** | **6.21e-06** | **5.75e-16** | **7.36e-14** | **2.71e-12** | **2.51e-10** | **4.47e-10** |
| refer2 vs refer2500 | **6.46e-06** | **3.50e-04** | **6.04e-06** | **7.39e-15** | **1.08e-14** | **5.08e-10** | **1.01e-10** | **5.59e-08** |
| refer2 vs refer5000 | **2.40e-06** | **3.42e-04** | **6.11e-06** | **1.13e-13** | **5.47e-15** | **3.57e-15** | **1.43e-15** | **2.62e-11** |
| refer2 vs refer7500 | **1.53e-06** | **3.53e-04** | **6.14e-06** | **6.99e-16** | **7.59e-12** | **9.73e-11** | **3.09e-10** | **1.98e-09** |
| refer2 vs refer10000 | **7.69e-06** | **3.42e-04** | **6.11e-06** | **2.45e-11** | **5.66e-17** | **2.63e-14** | **2.72e-14** | **4.85e-12** |
| refer10 vs refer100 | **1.71e-03** | **2.24e-04** | **5.99e-04** | **1.58e-04** | **4.44e-05** | **3.71e-05** | **1.64e-05** | **1.83e-05** |
| refer10 vs refer1000 | **3.33e-04** | **7.23e-05** | **3.90e-04** | **8.50e-05** | **2.47e-05** | **2.59e-05** | **1.33e-05** | **1.51e-05** |
| refer10 vs refer2500 | **3.23e-04** | **9.75e-05** | **3.54e-04** | **8.36e-05** | **3.11e-05** | **2.29e-05** | **2.04e-05** | **1.15e-05** |
| refer10 vs refer5000 | **1.66e-04** | **4.92e-05** | **3.65e-04** | **9.31e-05** | **2.86e-05** | **3.16e-05** | **2.65e-05** | **2.03e-05** |
| refer10 vs refer7500 | **1.81e-04** | **9.81e-05** | **3.70e-04** | **9.65e-05** | **2.70e-05** | **2.60e-05** | **1.77e-05** | **1.56e-05** |
| refer10 vs refer10000 | **7.58e-04** | **5.61e-05** | **3.64e-04** | **1.09e-04** | **4.14e-05** | **3.76e-05** | **3.01e-05** | **2.73e-05** |
| refer100 vs refer1000 | **2.88e-02** | **7.18e-04** | **1.00e-04** | **1.37e-04** | **1.10e-04** | **2.77e-04** | **1.26e-03** | **4.50e-04** |
| refer100 vs refer2500 | **1.70e-03** | **4.17e-04** | **1.78e-05** | **1.08e-04** | **1.92e-04** | **1.89e-04** | **1.49e-03** | **3.87e-04** |
| refer100 vs refer5000 | **2.50e-02** | **1.85e-03** | **2.43e-05** | **2.12e-04** | **1.70e-04** | **3.78e-04** | **1.99e-03** | **6.50e-04** |
| refer100 vs refer7500 | **4.87e-02** | **1.39e-03** | **4.27e-05** | **4.11e-04** | **8.38e-04** | **1.09e-03** | **3.96e-03** | **1.97e-03** |
| refer100 vs refer10000 | 5.08e-02 | **5.88e-04** | **2.51e-05** | **7.31e-04** | **7.31e-04** | **1.21e-03** | **4.03e-03** | **2.14e-03** |
| refer1000 vs refer2500 | 4.93e-01 | 2.70e-01 | 2.78e-01 | 6.41e-01 | 9.84e-01 | 5.97e-01 | 9.07e-01 | 8.31e-01 |
| refer1000 vs refer5000 | 1.85e-01 | 9.86e-01 | 1.14e-01 | 7.32e-01 | 7.58e-01 | 9.28e-01 | 8.53e-01 | 8.39e-01 |
| refer1000 vs refer7500 | 4.98e-01 | 2.87e-01 | 3.86e-01 | 1.90e-01 | **3.74e-02** | 9.23e-02 | 1.13e-01 | 1.12e-01 |
| refer1000 vs refer10000 | 4.55e-01 | 3.33e-01 | 1.15e-01 | 1.79e-01 | **4.96e-02** | 6.98e-02 | 1.05e-01 | 7.06e-02 |
| refer2500 vs refer5000 | 5.03e-02 | 2.92e-01 | 9.26e-01 | 8.44e-01 | 7.33e-01 | 6.04e-01 | 6.95e-01 | 6.82e-01 |
| refer2500 vs refer7500 | 1.55e-01 | 3.66e-01 | 7.53e-01 | 7.06e-02 | **2.85e-02** | **4.47e-02** | 6.60e-02 | 1.04e-01 |
| refer2500 vs refer10000 | 1.50e-01 | 9.86e-01 | 7.56e-01 | 5.12e-02 | **1.98e-02** | **3.31e-02** | **4.02e-02** | 8.05e-02 |
| refer5000 vs refer7500 | 5.21e-01 | 2.87e-01 | 6.05e-01 | 6.59e-02 | **4.95e-02** | 5.91e-02 | 8.63e-02 | 1.19e-01 |
| refer5000 vs refer10000 | 6.08e-01 | 3.53e-01 | 7.58e-01 | **2.25e-02** | 5.91e-02 | **2.73e-02** | **3.71e-02** | 5.61e-02 |
| refer7500 vs refer10000 | 9.22e-01 | 5.01e-02 | 4.89e-01 | 7.82e-01 | 4.31e-01 | 8.47e-01 | 7.67e-01 | 8.92e-01 |

Table 5: $p$-values of testing the statistical significance between two different types of source languages across various target games. We take $0.05$ as the threshold for our $p$-values, and bold all the significant results. For the comparison between {'refer1000', 'refer2500', 'refer5000'} and {'refer7500', 'refer10000'}, we also make the significant results be in red colour.

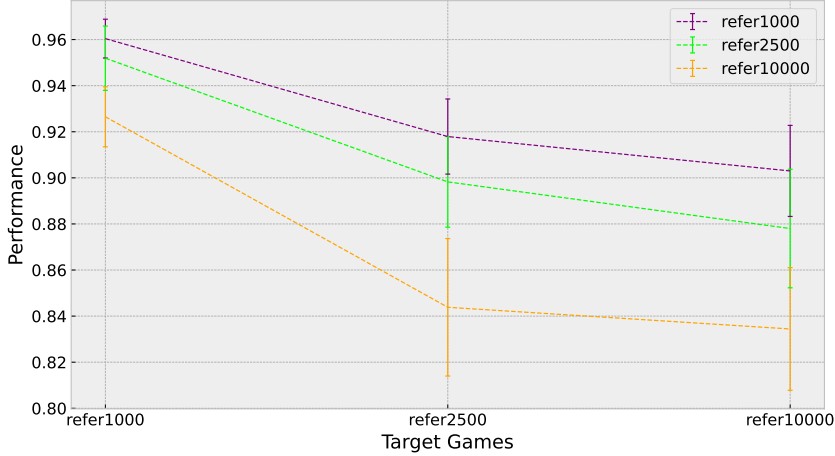

Figure 10: Results of language transfer with larger channel capacity. Error bars indicate the standard deviation of the mean values.

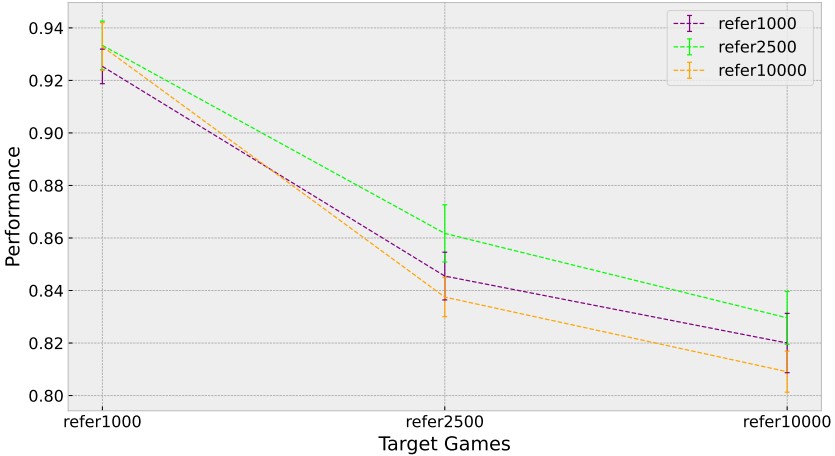

Figure 11: Results of language transfer with larger agent capacity. Error bars indicate the standard deviation of the mean values.

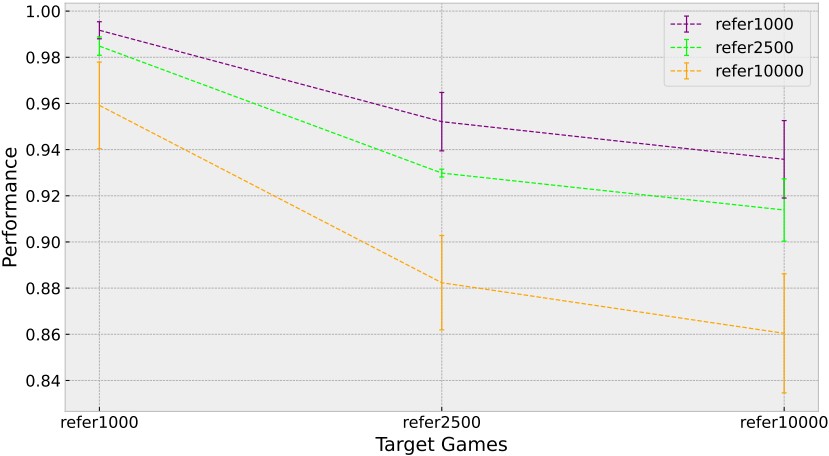

Figure 12: Results of language transfer with larger channel and agent capacity. Error bars indicate the standard deviation of the mean values.

