# OpenReview forum: "Expressivity of Emergent Languages is a Trade-off between Contextual Complexity and Unpredictability"
_ICLR.cc/2022/Conference — ICLR 2022 Poster_

### Official Review · Reviewer_jyzs · 2021-10-23

**Correctness:** 4
**Technical Novelty And Significance:** 3
**Empirical Novelty And Significance:** 3
**Recommendation:** 8
**Confidence:** 4

**Main Review:**

#### Strength
- This paper provides novel and interesting insights to the language emergence analysis by introducing the concepts of expressivity, context complexity and unpredictability. These concepts provide a deeper understanding of the behaviors of emergent languages from (referential) language games.

- The hypotheses are clearly stated, reasonable, and well supported by the experiment results.

- Besides the main results, the paper also proposes a simple update on training loss, contrastive loss, which improved the model behavior.


#### Weakness
- The introduced concepts (context complexity and unpredictability) are finally simplified to be only dependent on the candidate set size $|C|$. Also, experiments and analysis design are focused on $|C|$. The scope might be narrow (but also specific, so not necessarily a weakness).

- Although the hypotheses are clearly stated and supported, they are not very surprising based on intuition.

- Regarding the proposed training loss, to my understanding, it simply replaces the original candidate set with the batch. In this way, the contribution does not look significant.


#### Questions & Suggestions
- The definition of key concepts, context complexity and unpredictability, do not seem fully accurate. In text, the context complexity is defined as the expectation of the probability that "$C(x_t)$ includes an object from $N_k(x_t)$". However, equation 2 means the expectation of the probability that "a random object from $C(x_t)$ is in $N_k(x_t)$". Nonetheless, the simplified form $1-(\frac{|X|-|N_k(x_t)}{|X|})^{|C|}$ is according with the text definition. Maybe equation 2 needs to be modified. For unpredictability, the text definition is the expectation of "the probability that $C^{e+1}(x_t)$ contains an object that is not from $C^e(x_t)$", which seems wrong because this is clearly not monotonically decreasing on |C|. Based on the simplified form, I guess it should be the expectation of "the proportion of $C^{e+1}(x_t)$ that are not from $C^e(x_t)$".

**Summary Of The Paper:**

This paper studies the "expressivity" of emergent language in language games. To my understanding, "expressivity" is empirically the transferring ability of the language to unseen data. In this paper, the authors propose two factors of the underlying language game that can affect the expressivity of emergent language: context complexity and unpredicability. In referential games, the context is the distractor candidates for a sample. Complexity denotes how likely a "close" distractor will be included in the candidates. Unpredictability is how likely the context for a single sample will be different among epochs. This paper proposed hypotheses that both context complexity and unpredicability can improve language expressivity, but these two factors are "contradictory" and we need to have a trade-off between them. The hypotheses are supported by empirical experiment results.

**Summary Of The Review:**

This paper studies "expressivity" of emergent language, which is empirically the transferring ability of the language. The paper also defines "context complexity" and "unpredictability" of the underlying language game. The study shows that these two factors both contribute to the language expressivity, but they are contradictory on the candidate set size $|C|$ so a trade-off is needed. Generally, the analysis is novel and interesting. Although I have some concerns about the scope and significancy, the work still provides useful insights for language emergence study.

---

> ### Author Response · Authors · 2021-11-17
> **Responses to Reviewer jyzs**
>
> Thank you for identifying the strengths of this work.  We appreciate the thorough review, especially on identifying the specificity of our work. We believe all your concerns are addressed by the updates to the old version. If you have other concerns in mind, we would be happy to take them into consideration.
>
> > The introduced concepts (context complexity and unpredictability) are finally simplified to be only dependent on the candidate set size $|C|$. Also, experiments and analysis design are focused on $|C|$. The scope might be narrow (but also specific, so not necessarily a weakness).
>
> Since contextual complexity and unpredictability are somehow abstract, we defined them in a more specific way such that we could calculate their relations to the candidates set size. Without a specific definition of contextual complexity and unpredictability, we can only give some qualitative analysis of the relationships between context and expressivity, which we think is not as significant as a quantitative analysis.
>
> > Although the hypotheses are clearly stated and supported, they are not very surprising based on intuition.
>
> We believe the trade-off effect between complexity and unpredictability is not widely realised yet. Instead, a more popular belief is that more complex tasks could unilaterally lead to better expressivity. Other submissions to ICLR-2022, e.g. [1], still assume that the complexity of context is the only factor that influences the expressivity of emergent languages.
>
> References
> > [1] Emergent Communication at Scale, https://openreview.net/forum?id=AUGBfDIV9rL
>
> > Regarding the proposed training loss, to my understanding, it simply replaces the original candidate set with the batch. In this way, the contribution does not look significant.
>
> We updated the text about the contrastive loss to emphasize our contribution on the discovery of message type collapse and the mechanism by which the contrastive loss could help alleviate this problem.
>
> > The definition of key concepts, context complexity and unpredictability, do not seem fully accurate. In text, the context complexity is defined as the expectation of the probability that "$\mathcal{C}(x_t)$ includes an object from $\mathcal{N}_k(x_t)$". However, equation 2 means the expectation of the probability that "a random object from $\mathcal{C}(x_t)$ is in $\mathcal{N}_k(x_t)$". Nonetheless, the simplified form $1-\left(\frac{|X|-|\mathcal{N}_k(x_t)|}{|X|}\right)^{|C|}$ is according with the text definition. Maybe equation 2 needs to be modified. For unpredictability, the text definition is the expectation of "the probability that $\mathcal{C}^{e+1}(x_t)$ contains an object that is not from $\mathcal{C}^e(x_t)$", which seems wrong because this is clearly not monotonically decreasing on $|C|$. Based on the simplified form, I guess it should be the expectation of "the proportion of $\mathcal{C}^{e+1}(x_t)$ that are not from $\mathcal{C}^e(x_t)$".
>
> We replaced the indicator function in Equation 2 with the one that matches with our text description, and we replaced the text description of Equation 3 with the one you gave above. We also added a new appendix section(Appendix B in the new version) to further clarify the derivation of the solutions.

---

> > ### Comment · Reviewer_jyzs · 2021-11-19
> > **Reply to authors**
> >
> > Thank you for the reply and updates. I appreciate that the key definitions of complexity and unpredictability are clarified.
> >
> > Also, the point that "the trade-off effect between complexity and unpredictability is not widely realised yet" makes sense to me. Perhaps you can mention this explicitly in the paper (apologize if I missed it).
> >
> > For the proposed contrastive loss, it seems like a good empirical finding that it can improve the performance (by mitigating message type collapse), but I'm still not very convinced by its technical novelty.
> >
> > Overall, (after updating) the paper looks more sound and it provides inspiring insights to emergent language study. I would like to increase my score.

---

> > > ### Author Response · Authors · 2021-11-19
> > > **Reply to reviewer jyzs**
> > >
> > > Thank you for your consideration and acknowledgement of our efforts to address your reviews.
> > >
> > > We appreciate the note about the importance of using contrastive loss, and will rephrase the contribution in the paper. We’ll make it more clear that we pointed out that the contrastive loss could alleviate the message type collapse problem, which further influences the structure of the emergent languages (as discussed in Appendix C). We believe the use of this loss will support the emergent communication research, and it is our hope that this submission reinforces the benefits.
> > >
> > > We will also rephrase the tradeoff contribution in the paper to make it more clear and explicit.

---

### Official Review · Reviewer_27ev · 2021-11-01

**Correctness:** 2
**Technical Novelty And Significance:** 3
**Empirical Novelty And Significance:** 2
**Recommendation:** 3
**Confidence:** 4

**Main Review:**

The authors attempt to tackle the important issue of how functional pressures affect the properties of resulting emergent communication protocols. In particular, the are interested in quantifying the 'expressivity' of the resulting communication protocol, and give an operational definition of expressivity in terms of learnability by a randomly initialised listener. The authors note that mutual information is a poor measure for expressivity, and I like their proposed metric. It reminds me a little bit of various definitions of probing in the NLP literature, and perhaps a connection can be drawn here.

The resulting partial order in language transferability is described in text on page 8, but I think it would be better to draw the poset as a diagram, to make it easier to see what is comparable to what. In addition, I think the target games should be the x-axis of Figure 4, and the colours should be used for the source games. This way, we would be able to read off expressivity dominance immediately from the plot, as which curve was higher overall than the other. Further, it's unclear whether the generalisation performance is measured using a held-out set of language games, or whether the same games are used to train and evaluate the meta-suite of models.

The authors investigate the effect of two factors on language expressivity: scenario complexity and scenario unpredictability. Scenario complexity is defined as how many similar items each batch of data contains, on the assumption that items close in perceptual space are more difficult to distinguish. Scenario unpredictability is defined as the probability that the next batch of data contains an item not contained in the current batch.

I feel like complexity as the authors have defined is a very sensible metric. However, I'm not convinced by the unpredictability measure. It only considers bigram transitions, so considering a game with 4 inputs a, b, c, d and cycling the inputs endlessly as (a, b), (c, d) would be maximally unpredictable under this metric, when clearly it shouldn't be. I feel like the right notion of unpredictable should consider some kind of online learning scenario, such as prequential prediction. In addition, the human experiments used to motivate unpredictability rely on the fact that humans can remember past contexts when playing the game. In comparison, the models considered in the paper are memoryless, and the only effect of data presentation order is implicit in the optimisation dynamics. I understand that the intuition is that unpredictable environments force the speaker to be maximally informative in its utterances, but it's not clear that the proposed metric is the right way to capture this intuition.

Another contingent issue is that, at least as presented, complexity and informativeness are dependent variables, and the true controlled variable is batch size (indeed, the x axis of all the plots is batch size, and the formulae used to define complexity and unpredictability are not used anywhere else in the paper). This is because data points for a batch are sampled i.i.d. This means that the two axes of variation are not independent, which leads to conclusions which can't tease apart the effect of either measured variable, and reduces the overall impact of the paper as the empirical results suffer from confounding. I believe that alternative data sampling schemes would help tease apart the effect of both variables.

One final thing is that the 'contrastive' loss presented in equation 1 is contrasted with a 'referential' loss which is never explicitly given in the paper. To make the exposition self-contained, it would at least help to see how the 'referential' loss is computed, so that the reader can make a better comparison.

(As an aside: Wittgenstein did use the phrase 'language game', but his concern was not acquisition, but rather how meaning is modulated by context. The first sentence of the introduction therefore misrepresents his statements, and I feel should be removed.)

=============

Post author response:

> Memory of human participants also decay, thus the context is always limited in a finite number of time steps. Defining unpredictability on a longer history simply changes the power of Equation 3 in our setting, but doesn’t influence the growth relationship between unpredictability and context size. Therefore, it won’t influence our conclusion if we extend Equation 3 to a longer history.

This is only assuming a uniform bigram/trigram/n-gram distribution. Often in the real world, sequential data has long-term structure which isn't well captured by any n-gram transition model (which is why RNN/Transformer language models so dramatically outperform count-based ones), and so it's not obvious that using more sophisticated transition schemes would result in similar findings.

To be honest, I find the unpredictability measure more opaque after author response. Section 5.1 varies complexity while keeping unpredictability constant using something called a 'fixed batch', but it's not very well explained what this is, and why it makes minimises unpredictability. Again, I wonder why predictability of the future given the past is important for these communicating agents, considering they are memoryless across episodes. I brought up this point in my original review, and it is not addressed in the author response.

> We would be happy to include more useful definition of unpredictability if there’s other helpful definition you have in mind. We’ll also be happy to see more exploration on how unpredictability of context implicitly influence the training dynamics of agents.

I gave a definition: online learnability using prequential coding (see, e.g. The Description Length of Deep Learning Models, Blier and Ollivier 2018 for a description). There is a large literature relating unpredictability and online learning, and I believe that many ideas in that field will be applicable here.

> We would be happy to include a new data sampling scheme, if there’s some other useful method you have in mind.

Any kind of non-independent sampling would decouple complexity from unpredictability, such as stratified sampling of contexts according to similarity given a target. I believe that there is much scope to precisely control complexity and unpredictability using such schemes, which I believe would make a future revision of this paper much more solid.

After seeing the author resposne, I don't feel that many of the issues I raised in my review have been substantially addressed, and I have not changed my score. Most notably, the issue of the definition of unpredictability and how it affects the communication are still unclear, and I believe this needs to be clarified substantially in future revisions.

**Summary Of The Paper:**

The authors investigate what aspects of scenario design affect the resultant communication protocol in emergent communication. Specifically, the authors investigate whether scenario complexity (in terms of having many similar distractors) and scenario unpredictability (in terms of future batches not containing items to previous batches) affect the expressivity (measured in terms of learnability) of the resulting protocol.

Further, the authors show that defining a softmax loss over the entire batch of possible references (dubbed the 'contrastive' loss) outperforms the 'referential' loss used in previous works.

**Summary Of The Review:**

Some interesting ideas in this paper (measuring the effect of the environment on the effectiveness of the language protocol, introducing a new and sensible definition of effectiveness), but unfortunately I think there are issues outlined in the main review which preclude acceptance in the current state.

---

> ### Author Response · Authors · 2021-11-17
> **Response to Post Author Response (Part 2 of 2)**
>
> (Following Response to Post Author Response (Part 1 of 2) )
>
> 2. Why is it okay that we define unpredictability in terms of 1 step (e --> e+1) rather than  all of the previous trials, in the order the trial occurred.
>
> Like we said in our previous reply, no matter how many timesteps we use to define the unpredictability,  the unpredictability would still decrease as the size of candidate set increases. Therefore, our conclusion still holds with a different number of time steps.
>
> Beyond the above two responses, we would like to clarify some of the details of your comments and questions, as listed below.
>
> > This is only assuming a uniform bigram/trigram/n-gram distribution. Often in the real world, sequential data has long-term structure which isn't well captured by any n-gram transition model (which is why RNN/Transformer language models so dramatically outperform count-based ones), and so it's not obvious that using more sophisticated transition schemes would result in similar findings.
>
> We’d like to clarify some details of this question. In our paper, we assumed a uniform distribution over the inputs from the input space $\mathcal{X}$ and each input is a concatenation of one-hot vectors, as stated in Section 2.1.  According to your original comment, i.e. “ it only considers bigram transitions...”, the “n-gram” transitions refer to how the context for a given target varies across training epochs. However, in this sense, the description doesn’t match with the following discussion on RNN/Transformer language models which are designed to process inputs that are sequences. In our setting, the inputs are concatenations of one-hot vectors, not sequences, and thus changing the encoder of speakers to RNN/Transformer language models is not necessary.
>
> It seems that we need some further clarification about this question. Are you referring to the atomic elements in the communication game, or samples in the training batch, or something else? Your insights will allow us to more fully address this comment and update the final version of the paper.
>
> >To be honest, I find the unpredictability measure more opaque after author response. Section 5.1 varies complexity while keeping unpredictability constant using something called a 'fixed batch', but it's not very well explained what this is, and why it makes minimises unpredictability. Again, I wonder why predictability of the future given the past is important for these communicating agents, considering they are memoryless across episodes. I brought up this point in my original review, and it is not addressed in the author response.
>
> We address this comment in significant detail above. And, would like to point the reviewers and AC to the explanation of ‘fixed batch’ in the updated Appendix E.1, see in particular the definition of ‘referXf’.
>
> > Any kind of non-independent sampling would decouple complexity from unpredictability, such as stratified sampling of contexts according to similarity given a target. I believe that there is much scope to precisely control complexity and unpredictability using such schemes, which I believe would make a future revision of this paper much more solid.
>
> As we illustrated in the updates to Appendix B.3, our current experiment setup also controlled complexity and unpredictability separately, thus we believe that our results can support our conclusion. While we also agree that non-independent sampling is a reasonable extension for future research to better understand any relationship between complexity and unpredictability.
>
> >After seeing the author response, I don't feel that many of the issues I raised in my review have been substantially addressed, and I have not changed my score. Most notably, the issue of the definition of unpredictability and how it affects the communication are still unclear, and I believe this needs to be clarified substantially in future revisions.
>
> We have worked hard to update and clarify the definition of unpredictability in the paper, and in this discussion phase. We have also included more details on how unpredictability affects communication. We hope that you will take the substantial revision and discussion into account when considering and discussing the submission in the reviewer/AC discussion phase.

---

> > ### Author Response · Authors · 2021-11-17
> > **Response to Reviewer 27ev (Part 2 of 2)**
> >
> > > It reminds me a little bit of various definitions of probing in the NLP literature, and perhaps a connection can be drawn here.
> >
> > We are happy to include discussion connecting definitions of probing of models in NLP and our metrics of assessment for expressivity and learnability. We would be open to your suggestions of 1-2 references that we could incorporate into the submission.
> >
> > > The resulting partial order in language transferability is described in text on page 8, but I think it would be better to draw the poset as a diagram, to make it easier to see what is comparable to what.
> >
> > To emphasise that our Prediction 3 holds, we kept Figure 4 in our main body, and provide the comparison between the resulting language transferability in Appendix E. With the limited number of pages, we present  major conclusions in the main body of the text and include further explanation in the  appendices.
> >
> > > In addition, I think the target games should be the x-axis of Figure 4, and the colours should be used for the source games. This way, we would be able to read off expressivity dominance immediately from the plot, as which curve was higher overall than the other.
> >
> > The diagram described here is what is presented in Figure 9 we gave in Appendix E (Appendix C in the previous version). We kept Figure 4 in the main body since it that the peak of performance is always somewhere around refer1000-5000, which could better show our Prediction 3.
> >
> > > Further, it's unclear whether the generalisation performance is measured using a held-out set of language games, or whether the same games are used to train and evaluate the meta-suite of models.
> >
> > No, there’s no held-out set of language games in our transfer experiment. We use the same set of language games ($\mathcal{G}_s$ in Section 4.2) to train agents, and use the other set of language games ($\mathcal{G}_t$ in Section 4.2) to evaluate all emergent languages. More specifically, to compare the expressivity of two emergent languages, we would compare their generalisation performance across all language games in our target game set. For example, to compare 'refer1000’ and 'refer2500’, we would compare their generalisation performance on all of 'refer2’, 'refer10’, 'refer100’, 'refer1000’, 'refer2500’, 'refer5000’, 'refer7500’, 'refer10000’.
> >
> > > However, I'm not convinced by the unpredictability measure. It only considers bigram transitions, so considering a game with 4 inputs a, b, c, d and cycling the inputs endlessly as (a, b), (c, d) would $\dots$, but it's not clear that the proposed metric is the right way to capture this intuition.
> >
> > Memory of human participants also decay, thus the context is always limited in a finite number of time steps. Defining unpredictability on a longer history simply changes the power of Equation 3 in our setting, but doesn’t influence the growth relationship between unpredictability and context size. Therefore, it won’t influence our conclusion if we extend Equation 3 to a longer history.
> >
> > Regarding your concern about whether intuition is well captured, you’re correct that the example you gave above raises the maximal unpredictability with one-step assumption. But, it can be fixed by making longer history assumptions, and we showed above that this won’t change the growth relationship between unpredictability and context size, thus won’t influence our conclusion.
> >
> > We would be happy to include more useful definition of unpredictability if there’s other helpful definition you have in mind. We’ll also be happy to see more exploration on how unpredictability of context implicitly influence the training dynamics of agents.
> >
> >
> > > Another contingent issue is that, at least as presented, complexity and informativeness are dependent variables, and the true controlled variable is batch size ... I believe that alternative data sampling schemes would help tease apart the effect of both variables.
> >
> > The experiments in Section 5.1/5.2 unilaterally control complexity/unpredictability respectively. Although they are both dependent on candidate set size, we also showed that they can impact the expressivity separately. Besides, the batch size is not necessarily the same as the candidate set size, when using the standard referential loss function. Therefore, the true controlled variable is not only the candidates set size.
> >
> > We would be happy to include a new data sampling scheme, if there’s some other useful method you have in mind.
> >
> > > One final thing is that the 'contrastive' loss presented in equation 1 is contrasted with a 'referential' loss which is never explicitly given in the paper. To make the exposition self-contained, it would at least help to see how the 'referential' loss is computed, so that the reader can make a better comparison.
> >
> > We added a new appendix section (Appendix A in the new version) to give further comparison between the standard referential loss and the contrastive loss we used.

---

> > ### Author Response · Authors · 2021-11-24
> > **Response to Reviewer 27ev (Part 1 of 2)**
> >
> > Thank you for the thoughtful review and detailed assessment of the issues in the submission. We appreciate how you identified the sensibility of the way we define complexity, the value in measuring how the environment affects language effectiveness, and the connections to probing in NLP. We have addressed the issues that you outline in your main review, and hope that with these updates you would consider the contributions made in this work.
> >
> > > The first sentence of the introduction therefore misrepresents his statements, and I feel should be removed.
> >
> > We have rephrased the first sentence of the introduction to better represent Wittgenstein's position and incorporate Lewis' perspective, both of which motivate our current work on how meaning and language emerge.

---

> ### Author Response · Authors · 2021-11-24
> **Response to Post Author Response (Part 1 of 2)**
>
> Thanks for further clarifying some of your concerns. We paraphrase two of the larger important questions and provide further replies below. We also respond to the other questions at the end of this post. In the meantime, we are also hoping that you can provide more clarification about some of your questions as requested below.
>
> 1. Why is predictability of the future given the past is important for these communicating agents, considering they are memoryless across episodes?
>
> Predictability of the future given the past is important for these agents, even though they do not store information in an explicit memory module for use across training epochs (i.e. "memoryless across episodes"), because: 1) they store knowledge in their gradients and parameters, 2) because the data is sampled without replacement, 3) the data sampling and training dynamics will affect the learned representations.
>
> We gave in our common reply how our definition of unpredictability follows Winters et al. (2018), which leads us to a meaningful measure of unpredictability of context in our work in emergent communication.
>
> We appreciate that you are aligned with our intuition that “unpredictable environments force the speaker to be maximally informative in its utterances”, and we believe that our proposed metric does provide an informative perspective on this relationship.
>
> We agree with your intuition on the effect of predictability, i.e. there’s an implicit effect brought by the data presentation order in the training dynamics. In fact, introducing unpredictability (i.e. variance in the input and input context space) has also been commonly used in the computer vision community. For example, the preprocessing for images used in deep reinforcement learning tasks can include transformations such as `random shifts` to make the inputs vary across epochs (e.g Kostrikov et al. 2021). The unpredictability/variance of inputs are not explicitly measured by the online learning methods, but the effect of them still can improve the learned representation, which is shown by the improvement of performance (see Figure 1 from Kostrikov et al. 2021).
>
> To fully understand the implicit effect caused by the data presentation order covers much wider topics than emergent communication. That said,we look forward to exploring these questions in our  future work. In this work, we align with previous related work in cognitive science and proposed our definition of unpredictability in the context of emergent communication, as detailed in Section 3.1.
>
> Regarding the “online learnability using prequential coding” (Blier and Ollivier 2018), one challenge of this measure in our context is that it would conflate the learnability of emergent languages with the definition of unpredictability. Learnability is a connected research challenge in emergent communication, for example as investigated by Li and Bowling (2019).
>
> Since we focus on the effect from context on the expressivity of emergent languages, we opted to separate our definition of unpredictability from learnability which is again influenced by both communicating agents and emergent languages. In this work, we focus on the influence of the unpredictability of the contexts from epoch to epoch on expressivity of emergent languages. Similar to learnability, expressivity is also a specific property of emergent languages, which is learned  by agents through participation in language games. On the other hand, context is a part of the environment (i.e. part of the definition of the  language games), thus its properties and characteristics should be controlled by experimenters as best as possible to understand the  influence of  agents learning in these environments.
>
> Therefore, we chose a measure of unpredictability and complexity that can separate the effects from the outputs of the language games including the emergent languages and the trained communicating agents.
>
> >Reference:
> [1] Li, F., & Bowling, M. (2019). Ease-of-teaching and language structure from emergent communication. NeurIPS.
> [2] Kostrikov, I., Yarats, D., & Fergus, R. (2020). Image augmentation is all you need: Regularizing deep reinforcement learning from pixels. arXiv preprint arXiv:2004.13649.
>
> For the remaining discussion, please refer to the Response to Post Author Response (Part 2 of 2) below.

---

> ### Comment · Area_Chair_VbfJ · 2021-11-25
> **regarding unpredictability**
>
> Dear reviewer,
> Thanks for the review and discussions. I feel that too much of the discussion is focused on one point about how unpredictability should be defined. However, defining unpredictability does not seem to be the main result of this paper.
> It would be helpful to know what you think about the paper (the results and conclusions) assuming the current definition of unpredictability.

---

### Official Review · Reviewer_B1Wi · 2021-11-02

**Correctness:** 4
**Technical Novelty And Significance:** 2
**Empirical Novelty And Significance:** 4
**Recommendation:** 8
**Confidence:** 3

**Main Review:**

In my opinion, the biggest strength of the paper is the empirical evaluation conducted to support the claims. The experimental design and setup look sound. The definitions and different notions introduced in the paper seem natural and reasonable. At the same time, given the notions of complexity and unpredictability, a number of conclusions seem very natural based on intuition.

The paper is very well written and the arguments are backed up with reasonable experiments.

The proposed metric seems to work well in the given setting. Isn't the metric is dependent on the transfer learning abilities of the model to a certain extent? Will the partial ordering be consistent across different types of models?

The idea of applying contrastive loss is not particularly novel. It seems like to address the issue of space efficiency, they simply confined the previous method to batches rather than the entire set.

**Summary Of The Paper:**

The paper studies the properties of emergent languages in DL-based language games. In particular, they look at referential games where speakers emit messages and listeners need to identify the target object observed by the speaker from some set of candidate objects.

They define the expressivity of an emergent language as the amount of discriminatory information required to encode the inputs so that a listener can correctly decode them. In this setting, complexity refers to the similarity between different objects in a given context. In a context with higher complexity, more similar objects will be present, and more discriminatory information will have to be encoded so that a listener is capable of identifying the correct target. In their definition, the notion of unpredictability can be thought of as how stable the information necessary for encoding is across different trials.

Their primary contribution is to support the hypothesis that the expressivity of emergent language is determined by (and a trade-off between) the complexity and unpredictability of context in language games. The authors introduce a new measure to evaluate expressivity based on partial ordering between languages in terms of their generalization across tasks. They argue that mutual information is not the most appropriate measure to evaluate the expressivity of languages. They propose a contrastive loss which they show helps mitigate the issue of the collapse of message types.

**Summary Of The Review:**

Overall, it seems like a strong paper with a clear hypothesis and brings in a novel perspective which could be a worthwhile contribution in the subarea of emergent languages. I see no obvious issues with the paper.

---

> ### Author Response · Authors · 2021-11-17
> **Responses to Reviewer B1Wi**
>
> Thank you Reviewer B1Wi for identifying the strengths of this work and the contributions that we make to the field of language emergence. In particular, we appreciate your assessment of the soundness of our empirical evaluation setting.
>
> > Isn't the metric is dependent on the transfer learning abilities of the model to a certain extent? Will the partial ordering be consistent across different types of models?
>
> The metric is independent from the transfer learning abilities of models, but they are dependent on the games. For example, accuracy is not suitable for reconstruction games. If there are other useful metrics that you have in mind, we would be happy to include them.
>
> In Appendix F (in the new version), we also show that varying model architecture (with different hidden sizes) wouldn’t influence the partial ordering between the expressivity of emergent languages.
>
> For the metric we used in our work, we further clarify our reasons for choosing them in the updated Section 4.1. In short, since the purpose of emergent languages is to complete tasks, which is also the goal in the transfer learning framework, therefore both transfer learning and our work concern the performance across different tasks brought by the information conveyed in representation (or, embedding vectors) and emergent language respectively.

---

### Official Review · Reviewer_MdRY · 2021-11-02

**Correctness:** 4
**Technical Novelty And Significance:** 3
**Empirical Novelty And Significance:** Not applicable
**Recommendation:** 6
**Confidence:** 4

**Main Review:**

I like the overall idea of the paper on exploring the relationships between expressivity, complexity and unpredictability in emergent languages.

**Strengths:**

- The paper properly motivates with examples, the study of expressivity, complexity and unpredictability in the framework of deep learning based language games as a way to gain further insights into natural languages which I find really interesting.
- The paper introduces a novel measure of expressivity for emergent languages that is defined through its generalization performance across different tasks.
- The paper studies the usage of contrastive loss in referential games and this loss helps alleviate the collapse of message types in language games which is interesting and novel.
- The experiments are extensive which is great to see.
- Further discussions in the appendix is also nice. Particularly, those around mutual information.

**Weaknesses:**

I have included actions that can be taken to strengthen this paper. If these actions are addressed, I am willing to increase my score.

- **Section 3.1/4.1:**
    - **Complexity/unpredictability:** I don't fully understand what equations 2, 3 mean, both are of the form $f(x_t) = E_{x_t}[...]$. So, $x_t$ is both an argument to the function and the expectation. How is the expectation over $x_t$?
        - **Action:** Clarify equations 2 and 3.
    - **Complexity/unpredictability:** I am a little confused at how the closed form solutions for equations 2 and 3 were arrived at. Equation 2 for example, might be correct only if it is sampled with replacement, not without. Both are of the following form: the probability that a set A contains an object from set B where A, B are sampled from X without replacement. The solution to this is 1 - (|X| - |B|  \choose |A|) / (|X| \choose |A|).
        - **Action:** Correct or clarify solutions to equations 2 and 3.
- **Section 3.2:**
    - The paper does not fully motivate why Definition 3.1 is a good quantifiable metric for studying expressivity in emergent languages. How does generalization that is studied in emergent languages like [2] relate to this particular definition of expressivity?
        - **Action:** Compare expressivity with the notion of generalization studied in past works.
- **Section 4.1:**
    - Predictions in section 4.1 depends on the expressions being accurate in Section 3.1. It is also not clear to me how the definitions of unpredictability and complexity leads to predictions over expressivity.
        - **Action:** Clarify how we can expect these predictions to be true from the definitions provided.
- **Section 5.3:**
    - Looking at Figure 4 and the corresponding Table 4 (in the appendix), it does not seem that referY is statistically worse than referX for Y > X as claimed in this section.
        - **Action:** Including these comparisons with a statistical significance test would be needed to draw these conclusions.

**Comments:**

- **Section 2.2:**
    - What are source and target games? They are defined in a later section and pointing the readers to section 4.2 would make it easier to follow.
    - Section 4.2 is a little difficult to follow. This section defines ReferX and ReferX/f which is used in the rest of the paper. The definition for ReferX/f is particular is not very clear. The following statement is also not very clear: "since language games with fixed context are not good simulation of human communication , we keep only the ones having varying batches from $G_s$ in $G_t$."
        - **Action:** Include further discussions into the definitions for ReferX, ReferX/f, $G_s$ and $G_t$.

[1] Kirby, S., Tamariz, M., Cornish, H., & Smith, K. (2015). Compression and communication in the cultural evolution of linguistic structure. *Cognition*, *141*, 87-102.

[2] Chaabouni, R., Kharitonov, E., Bouchacourt, D., Dupoux, E., & Baroni, M. (2020). Compositionality and generalization in emergent languages. *arXiv preprint arXiv:2004.09124*.

**Summary Of The Paper:**

This paper studies the expressivity, complexity and unpredictability of emergent languages in referential games. The authors demonstrate that the expressivity of emergent languages is a trade-off between the complexity and unpredictability of the context that the languages are used in. They also introduce a contrastive loss based training method for referential games that alleviates the collapse of message types often seen when using other standard training methods.

**Summary Of The Review:**

The paper properly motivates the study of expressivity, complexity and unpredictability in the framework of deep learning based language games as a way to gain further insights into natural languages. The paper also studies the usage of contrastive loss in referential games and this loss helps alleviate the collapse of message types in language games. The key contributions of the paper is predicated on the definition and closed form expressions for unpredictability and complexity. However, I do not believe that those expressions are sound. Although, there are extensive experiments, some conclusions are not well supported. Therefore, as it stands, I would not recommend the paper to be accepted.

---

> ### Author Response · Authors · 2021-11-17
> **Responses to Reviewer MdRY**
>
> Thank you for identifying the strengths of this work. We appreciate the thorough review, and clear actions to strengthen the paper. We have addressed each action systematically as described. We believe that the expressions are made more sound and the conclusions are more well supported through these updates. If you have other actions in mind, we would be happy to take them into consideration.
>
> > Clarify equations 2 and 3, and correct/clarify solutions to equations 2 and 3.
>
> We have corrected the errors, and added more details about deriving the solutions in a new appendix section (Appendix B in the new version).
>
> > Is $x_t$ both an argument to the function and the expectation. How is the expectation over $x_t$?
>
> Yes, we defined the context for “a given $x_t$”, thus both complexity and unpredictability are also dependent on $x_t$. Therefore, the definition we gave in Section 3.1 are all expectations over $x_t$.
>
> > Compare expressivity with the notion of generalization studied in past works.
>
> We updated Section 3.2 to better motivate the definition of expressivity based on generalisation performance.
>
> Regarding the notion of generalisation, the authors of [2] evaluate the generalisation performance on a single kind of target game, while we use generalisation performance on a set of games as a metric for the expressivity of emergent languages.
>
> At the same time, [2] shows that our input space (containing 10,000 samples) is large enough for agents to achieve a reasonable generalisation performance (>99%). We use this as evidence to support the set-up of our language games.
>
> Referecne:
> > [2] Chaabouni, R., Kharitonov, E., Bouchacourt, D., Dupoux, E., & Baroni, M. (2020). Compositionality and generalization in emergent languages. arXiv preprint arXiv:2004.09124.
>
> > Clarify how we can expect these predictions to be true from the definitions provided.
>
> We updated Section 4.1 to give better derivation of the three predictions. More specifically, we show that the first two predictions directly follow the existing findings from other domains, and there’s naturally a tradeoff since we cannot increase both complexity and unpredictability at the same time.
>
> > Including these comparisons with a statistical significance test would be needed to draw these conclusions.
>
> We have included these comparisons supporting Table 4 and Fig 4 with statistical significance testing (see Table 5, Appendix E) and have summarized when refer7500/refer10000 is statistically worse (p<0.05) than refer1000/2500/5000 on some target games, thus the expressivity of refer1000/2500/5000 is better.
>
> > What are source and target games? They are defined in a later section and pointing the readers to section 4.2 would make it easier to follow.
>
> We updated both the last paragraphs of Section 2 and the first paragraph of Section 4.2 to address this clarity concern. In short, source games are the games where agents can negotiate emergent communication protocols in order to complete tasks. Target games are the games where we train listeners to complete tasks with only the messages from an emergent language.
>
> > Include further discussions into the definitions presented in Section 4.2.
>
> We have clarified the ambiguous wording and definitions in Section 4.2. Further discussions into the definitions for "referX", "referXf", $\mathcal{G}_s$, and $\mathcal{G}_t$ are added in Appendix E.

---

> > ### Comment · Reviewer_MdRY · 2021-11-22
> > **Reply to authors**
> >
> > The changes made to the paper have definitely strengthened it. I particularly appreciate the clarifications and inclusion of significance tests to the paper. The experimental rigor is commendable. I think the paper now makes a valuable contribution to the emergent communication community.
> > As such, I have increased my score and would recommend the paper to be accepted.

---

> > > ### Author Response · Authors · 2021-12-02
> > > **Reply to Reviewer MdRY**
> > >
> > > Thank you for your recommendation and acknowledgement of our efforts to address your concerns.
> > >
> > > We appreciate the note about the novelty of the expressivity concept in emergent languages as well as the rigour of our experiment design. Thanks again for the valuable suggestions for improving the quality of our paper.

---

### Author Response · Authors · 2021-11-17
**Overall response**

We want to thank the reviewers for their time, constructive feedback, and diligent reviews. Please find responses to your comments and questions below. We have made updates to the paper, which are summarized below in a list of Major Updates and then described in detail how the updates address the reviewers’ comments.

We summarize the Major Updates to the paper as follows:

1. We updated the last part of both abstract and introduction sections to highlight our contributions on the discovery of the problem of message type collapse.
2. We added a new appendix section (Appendix A) to clearly describe connections between the standard referential loss function and the contrastive loss function used in our work.
3. We corrected Equation 2 and 3, and added a new appendix section (Appendix B) to further clarify solutions to both of them.
4. We updated the second paragraph in Section 3.2 to better motivate our Definition 3.1 following the inspirations from the transfer learning community.
5. We updated Section 4.1 to better clarify how we derived the three predictions following our hypothesis and definition of context.
6. We added exhaustive details on statistical significance testing to support the results in the language transfer experiment (Section 5.3) in Appendix E.
7. We added more details about the naming of games and languages we used in the language transfer experiment in Appendix E.

---

We’ll also add the following paragraphs to Appendix B.3 to explain how our measure of unpredictability is analogous to the human experiments described in Winter et al. (2018).

In this section, we first derive our measure of unpredictability following Winter et al. (2018).
In their experiments, the input space consists of $16$ referents. There are $2$ different attributes, *colour* and *shape*, and each attribute has $4$ different values.

By setting the size of candidate set to $4$, they define the following two types of context:

 * Case 1: **shape-different** where all the $3$ different distractors have identical colour to the target, but different shapes. Since there are only $4$ shapes and candidate set size is also $4$, in this setting, the distractors for a given target will never change during the games. This is analogous to our ``fixed-batch'' games which we define and provide illustrative examples of in Appendix E.1. The partition of the context in the fixed-batch games remains identical from epoch (trial) to epoch (trial). The fixed-batch setting minimizes unpredictability by eliminating the variance of the distractors in the full context for a given target.

 * Case 2: **mixed** where all the $3$ different distractors are sampled uniformly at random from the whole input space without replacement. Thus, it is more likely that the distractors have different shapes as well as different colours to the target. In the **mixed** case, the distractors for a given target vary across different trials during the games.

In Case 1, the distractors for a given target do not change each trial. In Case 2, the distractors do change each trial. Thus, it is more likely that a context changes in the **mixed** setting.

Each trial in the human experiments of Winters et al. (2018) corresponds to a training epoch in our deep learning setting. They described how the **mixed** context has higher unpredictability (Section 1.4 in (Winters et al., 2018)), and our sampling procedure matches theirs, namely Bernoulli sampling without replacement from trial to trial.

This is analogous to our experimental setting where higher unpredictability corresponds to an increased likelihood that a given target has a context with different distractors across training epochs. This further leads us to our definition of 'the unpredictability of context' as defined in Equation 3.

Note that there is also complexity (under our definition) involved in the above two types of context. We focus only on the variability part of the context in our definition of unpredictability since the complexity part has been taken into consideration by the neighbourhood set defined in our Section 3.1.

> Reference:
[1] James Winters, Simon Kirby, and Kenny Smith. Contextual predictability shapes signal autonomy. Cognition, 176:15–30, 2018.

---

### Decision · Program_Chairs · 2022-01-20

**Decision:**

Accept (Poster)

**Comment:**

This paper studies the expressivity, complexity and unpredictability of emergent languages in referential games. The authors defined measures of complexity and unpredictability and empirically showed that the expressivity of emergent languages is a trade-off between the complexity and unpredictability of the context that the languages are used in. They introduced a contrastive loss based training method that alleviates the collapse of message types seen using standard referential loss functions.

The paper is controversial among the reviewers. On the positive side, most liked how the paper has a clearly stated hypothesis and extensive evaluations which makes a clear contribution to the field of emergent languages. On the negative side, the paper only shows the results in an artificial setting where the key variables are highly simplified (e.g. size of candidates). The main negative review argue the authors used an inappropriate definition of unpredictability and that the batch size is actually the key independent variable instead of what is claimed. The paper does somewhat equate batch size with the candidate size that is so important to their results (after eq (1)), but they seem to measure candidate size in the key figures. Perhaps an experiment controlling for batch size independently of the candidates size can address this issue. On the point of defining unpredictability, the other reviewers and I find the given definition to be reasonable and at least defensible. However, the reviewer remained unconvinced. More generally, the paper relies on one definition of the concepts measured in one setting to make a general claim, which is at risk of missing other important variables. Overall, most reviewers found the scope to be sufficient, and two improved their scores after the discussion.

Recommendation: accept